# Universality of closed nested paths in two-dimensional percolation

Yu-Feng Song [1,2,⋆], Jesper Lykke Jacobsen [3,4,5,†] Bernard Nienhuis[6,‡], Andrea Sportiello[7,∘] and Youjin Deng[1,2,§]

**1** Hefei National Laboratory for Physical Sciences at Microscale and Department of Modern Physics, University of Science and Technology of China, Hefei, Anhui 230026, China
**2** MinJiang Collaborative Center for Theoretical Physics, Department of Physics and Electronic Information Engineering, MinJiang University, Fuzhou, Fujian 350108, China
**3** Laboratoire de Physique de l'École Normale Supérieure, ENS, Université PSL, CNRS, Sorbonne Université, Université de Paris, F-75005 Paris, France
**4** Sorbonne Université, École Normale Supérieure, CNRS, Laboratoire de Physique (LPENS), F-75005 Paris, France
**5** Université Paris Saclay, CNRS, CEA, Institut de Physique Théorique, F-91191 Gif-sur-Yvette, France
**6** Delta Institute of Theoretical Physics, Instituut Lorentz, Leiden University, P.O. Box 9506, NL-2300 RA Leiden, The Netherlands
**7** LIPN, Université Sorbonne Paris Nord, CNRS, F-93430 Villetaneuse, France

⋆ jlyfsong@mail.ustc.edu.cn , † jesper.jacobsen@ens.fr , ‡ nienhuis@lorentz.leidenuniv.nl , ∘ andrea.sportiello@lipn.univ-paris13.fr , § yjdeng@ustc.edu.cn

## Abstract

Recent work on percolation in $d = 2$ [J. Phys. A 55 204002] introduced an operator that gives a weight $k^\ell$ to configurations with $\ell$ 'nested paths' (NP), i.e. disjoint cycles surrounding the origin, if there exists a cluster that percolates to the boundary of a disc of radius $L$, and weight zero otherwise. It was found that $\mathbb{E}(k^\ell) \sim L^{-X_{\rm NP}(k)}$, and a formula for $X_{\rm NP}(k)$ was conjectured. Here we derive an exact result for $X_{\rm NP}(k)$, valid for $k \geq -1$, replacing the previous conjecture. We find that the probability distribution $\mathbb{P}_\ell(L)$ scales as $L^{-1/4}(\ln L)^\ell[(1/\ell!)\Lambda^\ell]$ when $\ell \geq 0$ and $L \gg 1$, with $\Lambda = 1/\sqrt{3}\pi$. Extensive simulations for various critical percolation models confirm our theoretical predictions and support the universality of the NP observables.

---

## Contents

---

## 1 Introduction

After more than 60 years of intensive study since 1957, percolation [1–5] still remains a central and active research topic in statistical mechanics and probability theory [6–8]. It is a prototypical and perhaps the simplest example of collective behavior. For *bond percolation*, each lattice edge or bond is independently occupied with probability $p$, or left vacant (empty). Two sites are said to be connected if there is a path of occupied bonds from one site to the other, in which each pair of subsequent bonds is adjacent to a common site. A cluster is a maximal set of sites connected to each other, and accordingly the set of all lattice sites can be decomposed

into clusters (including clusters of just one isolated site). For *site percolation*, each lattice site is independently occupied with probability $p$, or left vacant, and any pair of neighboring occupied sites is said to be connected. A cluster is then a maximal set of mutually connected occupied sites, and the set of occupied sites is partitioned into clusters.

Clusters are small for small $p$, but letting $p$ tend to the percolation threshold $p_c > 0$ from below, $p \uparrow p_c$, causes the emergence of a so-called giant cluster at $p > p_c$, namely a cluster that occupies a finite fraction of the lattice sites, in the thermodynamic limit. The percolation transition is one of the simplest examples of a continuous phase transition [9,10] and provides a vivid illustration of many important concepts of critical phenomena [11]. In the sub-critical phase ($p < p_c$) the correlation length $\xi$, a scale proportional to the diameter of the largest (finite) cluster, diverges as $\xi \sim (p_c - p)^{-\nu}$ when $p \uparrow p_c$, with $\nu$ the correlation-length exponent. In the super-critical phase ($p > p_c$), the probability $m$ that a randomly chosen site is in the giant cluster vanishes as $m \sim (p - p_c)^{\beta}$ when $p \downarrow p_c$. In many textbooks, it is claimed that $\nu$ and $\beta$ are essentially the only two basic independent exponents, from which other critical exponents can be obtained via (hyper-)scaling relations.

Exact calculations of $\nu$ and $\beta$ are available for the Bethe lattice (or Cayley tree), and for the complete graph, both of which can be considered as the limit of infinite spatial dimension [2]. Furthermore, these mean-field results, $\nu = 3/d$ and $\beta = 1$, are believed to hold already for any dimension $d \geq d_u$ above the upper critical dimension, $d_u = 6$ [12,13]. In two dimensions (2D), exact values, $\nu = 4/3$ and $\beta = 5/36$, were predicted by the Coulomb-gas (CG) method [14], conformal field theory [15] and stochastic Loewner evolution (SLE) techniques [16], and crowned by a rigorous proof for triangular-lattice site percolation [17]. For $2 < d < d_u$, estimates of $\nu$ and $\beta$ are available from numerical simulations and perturbative methods.

## 1.1 General considerations

**Fractal structures .**    At the percolation threshold $p_c$, clusters are scale-invariant, with fractal dimension $d_F = d - \beta/\nu$. To further characterize geometric structures of critical percolation clusters, one considers also the fractal dimensions of geometrical objects other than the giant cluster itself, for instance the set of red (or pivotal) bonds, backbones, shortest paths, hulls and external perimeters [2,18]. The red-bond dimension is $d_R = 1/\nu$, whereas the other exponents are considered to be independent of $\nu$ and $\beta$, at odds with the over-simplified textbook scenario mentioned above. In 2D some exact results are known, including $d_F = 91/48$ for clusters, $d_H = 7/4$ for hulls and $d_E = 4/3$ for external perimeters [2,20]. Very recently the value of $d_B$ for backbones was determined [21] using SLE and turns out to be transcendental. But despite many efforts, the exact value of $d_S$ for shortest paths is still unknown. For $d \geq 6$, one has $d_F = 2d/3$ and $d_B = d_S = d_R = d/3$ [12,13]. For $2 < d < 6$, only numerical estimates are available.

**Correlation functions .**    It is also well known that, at $p_c$, a variety of connectivity probabilities between two far-away regions decay algebraically with distance $r$ as $r^{-2X}$, where $X$ is called the scaling dimension [2,18]. Alternatively, one can consider a domain with the topology of a disc. The corresponding one-point function, giving the probability that the chosen connectivity exists between the center of the disc and its boundary, then decays with the disc radius $r$ as $r^{-X}$. The property 'connecting the center of a disc to its boundary' we will henceforth indicate with the word *radial*. The typical example of such connectivity observables concerns the so-called magnetic operator, which gives the probability that the two different regions belong to the same cluster. The corresponding exponent is $X = X_F = d - d_F$ determining the fractal dimension $d_F$ of the percolating cluster. In two dimensions $X_F = 5/48$. In the following we discuss a number of generalizations of the magnetic operator, culminating with the nested-path operator which is the focus of this work.

**Duality.** In 2D, dual or empty clusters and paths can be related to the corresponding construction for empty elements by duality [20]. When the distinction between dual clusters and the original clusters needs to be emphasized, we shall call the original clusters 'primal'. For bond percolation, clusters for empty elements consist of bonds on the dual lattice, where a dual bond is occupied iff the intersecting original bond is empty, and vice versa. For site percolation, dual or 'empty' clusters consist of connected empty sites on the matching lattice. For the definition of dual and matching lattices, see Refs. [3,5]. In 2D at $p_c$ both direct clusters and dual clusters are fractal. For self-dual and self-matching lattices (for bond– and site percolation respectively), the clusters and dual clusters have the same properties, so that the boundaries between them are symmetric. It follows that $p_c = 1/2$ for such lattices.

## 1.2  Operators and exponents

**Monochromatic $N$-arm operator.** A direct generalization of the magnetic operator is the family of monochromatic $N$-arm (MA) operators, defined for integer $N \geq 1$. The two-point function of the MA operator is defined as the probability that two distant small regions are connected by at least $N$ independent paths in the same cluster. Two paths are called independent if they do not share a common occupied bond (site) for bond (site) percolation, and do not cross [22, 23]. The one-point function of this observable is the probability that the cluster that contains the center of a disc with radius $r$ contains $N$ independent radial paths. The corresponding exponent is denoted as $X_{MA}(N)$, and, for $N = 1$, it reduces to the magnetic exponent. The $N = 2$ case is called the backbone exponent $X_{MA}(2) = d - d_B$, of which the exact value remained a challenge until very recently. Nolin et al. [21] successfully determined the value of $X_{MA}(2)$ as the root of a transcendental equation, with a value in good agreement with the best numerical estimates [19, 25]. This striking result provides an example that the 2D critical exponents do not necessarily take fractional values. Exact values of $X_{MA}(N)$ are still unavailable for $N \geq 3$.

**Polychromatic $N$-arm operator.** Besides the monochromatic $N$-arm operator, also the polychromatic $N$-arm (PA) operator is an object of study. The corresponding exponent $X_{PA}(N)$ governs the probability that two patches are connected by $N$ paths of which some are on primal clusters, and others are on dual clusters. Remarkably, $X_{PA}(N)$ is equal to the 'watermelon' exponent, $X_{WM}(N)$, to be introduced next. This equality was first argued succinctly in Ref. [20]. Below, in Sec. 2.2, we present a more detailed version of the argument.

**Watermelon operator.** The $N$-arm watermelon exponent [26, 27] governs the probability that two distant patches are connected by $N$ cluster boundaries (the two-point function) or that there are $N$ radial cluster boundaries (the one-point function). Its value is known to be

$$X_{WM}(N) = \frac{N^2}{12} - \frac{1}{12} \,. \tag{1}$$

For $N = 2$, the two-point function gives the probability that two points sit on the hull of the same cluster, so that $X_{WM}(2) = d - d_H = 1/4$. An observer passing around the insertion point of an $N$-arm watermelon operator once, crosses $N$ cluster boudaries. Thus, for odd $N$ the cluster he started in must have switched from empty to occupied or vice versa. Thus the operator requires anti-cyclic conditions (empty $\leftrightarrow$ occupied) under a full rotation around its insertion point. This is analoguous to the well-known disorder operator of the Ising model. A more detailed description will be given in Sec. 2.2.

For even $N$, Eq. (1) governs the decay of the probability that two distant regions are connected by $N/2$ distinct clusters, in which each cluster corresponds to two boundaries. In other

130   words, the $N$-cluster correlation functions in 2D have the scaling dimension $X = N^2/3 - 1/12$.
131   Further, a refined family of $N$-cluster correlations can be constructed according to the require-
132   ments of logarithmical conformal field theory and the relevant symmetric group, and such a
133   construction is valid both in 2D and higher dimensions [24, 28–31]. In 2D, the exact values
134   of these $N$-cluster exponents can be inferred from the branching rules of the symmetric group
135   down to the cyclic group and from exact CFT results [24].

136   **Nested-loop operator.**    There exists another family of operators based on cluster bound-
137   aries, called nested-loop (NL) operator [32, 33]. To describe it we again consider the one-point
138   function on a domain with the topology of a disc, of linear size (diameter) $L$. For each config-
139   uration, let $\ell$ denote the number of cluster boundaries surrounding the center of the domain.
140   The NL operator assigns a statistical weight, $k \in \mathbb{R}$, to each of these boundaries. Then the
141   one-point correlator, $W_{\mathrm{NL}}(k) \equiv \langle k^\ell \rangle$, is parametrized by $k$. This correlator $W_{\mathrm{NL}}(k)$ varies with
142   $L$ as $L^{-X_{\mathrm{NL}}(k)}$ at criticality. By CG and CFT methods, the exponent $X_{\mathrm{NL}}$ is found to be

$$X_{\mathrm{NL}}(k) = \frac{3}{4}\phi^2 - \frac{1}{12}\,, \qquad k = 2\cos(\pi\phi) \geq -2\,. \tag{2}$$

143   For $-2 \leq k \leq 2$, $\phi$ is real, while for $k > 2$ it is purely imaginary. The name 'nested loop'
144   refers to the fact that the relevant cluster boundaries are closed and must be nested, as they
145   do not cross each other. Some special cases are the following. For $(k, \phi) = (1, 1/3)$, the
146   weights of the configurations are unaffected by the insertion of the NL operator, implying
147   $W_{\mathrm{NL}}(1) = 1$, and $X_{\mathrm{NL}}(1) = 0$. For $(k, \phi) = (0, 1/2)$, $W_{\mathrm{NL}}(k)$ corresponds to the probability that
148   $\ell = 0$. When $\ell = 0$ the cluster containing the center is connected to the boundary. Thus,
149   $X_{\mathrm{NL}}(0) = X_{\mathrm{F}} = 5/48$, the magnetic scaling dimension.

150   **Nested-path operator.**    In a recent article [34], we introduced what we call the nested-path
151   (NP) operator, whose definition draws on several of the developments outlined above. It is the
152   main object of study also in this article. The watermelon (WM) operator and the nested-loop
153   (NL) operator are both defined in terms of cluster boundaries, emanating from the insertion
154   point or surrounding it, respectively. One can consider paths over clusters in the same two
155   topologies. While the monochromatic $N$-arm operator (MA) measures the probability that $N$
156   paths emanate from an insertion point, it is naturally complemented with an operator that
157   weights the monochromatic closed paths nesting around the insertion point. Like the $N$-arm
158   operator, we may distinguish two varieties: a monochromatic case where all paths are on the
159   primal cluster, and a polychromatic one with some paths on primal and some on dual clusters.
160   Where the distinction is important we will refer to the monochromatic nested-path (MNP)
161   operator and the polychromatic nested-path (PNP) operator, while the label NP is used for
162   both.

163   We define the NP operators as follows. Let $\ell$ be the maximum number of independent
164   nested closed paths surrounding the center that can be drawn on primal and dual clusters.
165   Further, let $\mathcal{R}$ be the indicator function that there exists a radial cluster. We then define the
166   continuous families of NP correlators as $W_{\mathrm{MNP}}(k) \equiv \langle \mathcal{R} \cdot k^\ell \rangle$, and $W_{\mathrm{PNP}}(k) \equiv \langle k^\ell \rangle$. This assigns
167   in both cases a statistical weight $k \in \mathbb{R}$ to each independent closed path (analogously to the NL
168   operator), while for the MNP operator only the configurations with $\mathcal{R} = 1$ contribute. Notice
169   that the factor $\mathcal{R}$ ensures that all the surrounding paths (if any) are contained in the same
170   percolating cluster, and if $\ell > 0$ the percolating cluster must be unique. This guarantees that
171   the nested paths measured by $W_{\mathrm{MNP}}(k)$ are monochromatic. The one-point functions $W_{\mathrm{MNP}}(k)$
172   and $W_{\mathrm{PNP}}(k)$ vary with domain diameter $L$ as $L^{-X_{\mathrm{MNP}}(k)}$ and $L^{-X_{\mathrm{PNP}}(k)}$ respectively, thus defining
173   the exponents $X_{\mathrm{MNP}}$ and $X_{\mathrm{PNP}}$.

174   For two special values of $k$, the NP correlators can be readily inferred. First, $W_{\mathrm{MNP}}(1)$
175   reduces to the percolating probability $\langle \mathcal{R} \rangle$, which is known to decay as $L^{-X_{\mathrm{F}}}$. This implies

$X_{\text{MNP}}(1) = X_{\text{F}} = 5/48$. More trivially, $W_{\text{PNP}}(1) = 1$ implies $X_{\text{PNP}}(1) = 0$. Second, the configurations contributing to $W_{\text{MNP}}(0)$ have a primal radial path, because of the factor $\mathcal{R}$, and since they have no primal path surrounding the center, they must also have a dual radial path. Likewise since the configurations contributing to $W_{\text{PNP}}(0)$ have neither a primal path nor a dual path surrounding the center, they must have both a dual and a primal radial path. This implies the existence of two radial cluster boundaries. The dominant contributions to $W_{\text{PNP}}(0)$ and $W_{\text{MNP}}(0)$ are thus those of the $N = 2$ path watermelon operator, implying $X_{\text{PNP}}(0) = X_{\text{MNP}}(0) = X_{\text{WM}}(2) = 1/4$. Furthermore, in Ref. [34] we proved the identity $W_{\text{MNP}}(2) = 1$ for site percolation on regular or irregular planar triangulation graphs of any size $L$, and for any shape and position of the center. By universality we infer $X_{\text{MNP}}(2) = 0$ for site or bond percolation on any 2D lattice.

In [34] we only considered the monochromatic nested paths, so the label NP in that paper corresponds to MNP here. There we conjectured an analytical formula for $X_{\text{MNP}}(k)$, as a function of $k$, on the basis of numerical results. Under the parametrization $k = 2\cos(\pi\phi)$, this conjecture reads $X_{\text{MNP}}(k) = (3/4)\phi^2 - (5/48)\phi^2/(\phi^2 - 2/3)$. It reproduces the known exact results for $k = 0, 1, 2$ and agrees very well with numerical estimates of $X_{\text{MNP}}(k)$ for other values of $k$. Unfortunately, this formula turns out to be incorrect. Below in Sec. 2.3, we shall provide a rigorous argument which relates the one-point functions $W_{\text{MNP}}(k)$ and $W_{\text{PNP}}(k)$ to the one-point NL function $W_{\text{NL}}(k')$ where the weight of the loop $k'$ has a simple relation to the weight $k$ of the nested paths. In view of Eq. (2) this leads to the explicit expression for the MNPs:

$$X_{\text{MNP}}(k) = \frac{3}{4}\phi^2 - \frac{1}{12}, \qquad k = 1 + 2\cos(\pi\phi), \tag{3}$$

For the polychromatic case the expression in terms of $\phi$ is the same, but its relation to the NP weight is different:

$$X_{\text{PNP}}(k) = \frac{3}{4}\phi^2 - \frac{1}{12}, \qquad k = \frac{1}{2} + \cos(\pi\phi), \tag{4}$$

Obviously, these expressions reproduce the known exact results mentioned above.

## 1.3 Outline and overview

The main purpose of this work is two-fold: to derive theoretically the correct analytical formulae (3) and (4) for the NP exponents, and to examine their universality. To this end we perform extensive Monte Carlo (MC) simulations for a number of critical percolation models, including one bond- and five site-percolation systems, and study an extended set of quantities. The universality of the power-law scaling for the one-point MNP function is well demonstrated and the estimates of the MNP exponent agree well with the derived formulae.

In addition, we study the probability distribution $\mathbb{P}_\ell(L)$ that the cluster percolates from the center site to the boundary ($\mathcal{R} = 1$) and supports $\ell$ nested paths. Since the analysis of $\mathbb{P}_\ell(L)$ as well as the MC study concerns only the monochromatic nested paths, we omit in the relevant sections the corresponding label MNP, and replace $W_{\text{MNP}}(k)$ by $\mathcal{W}_k$, or with explicit dependence on the system size, $\mathcal{W}_k(L)$. Likewise $X_{\text{MNP}}$ will be simply denoted by $X$. For $\ell = 0$, notice that, by definition, $\mathbb{P}_0 \equiv W_{\text{MNP}}(0) \sim L^{-1/4}$. For each $\ell \geq 1$, on the basis of formula (3) we show that the leading scaling behavior of $\mathbb{P}_\ell(L)$ is $L^{-1/4}(\ln L)^\ell[(1/\ell!)\Lambda^\ell]$, with $\Lambda = 1/\sqrt{3}\pi$. We then consider the average number of nested paths conditioned by the existence of a percolating cluster, $N \equiv \langle \ell \cdot \mathcal{R} \rangle / \langle \mathcal{R} \rangle$. It is shown that, as $L$ increases, this conditional path number diverges logarithmically as $N \simeq \kappa \ln L$, with $\kappa = 3/8\pi$. The theoretical predictions for $\mathbb{P}_\ell$ and $N$ are well confirmed by our high-precision MC results.

The remainder of this work is organized as follows. Section 2 demonstrates relations between the polychromatic $N$-arm and the watermelon exponent and between the NP operator and the NL operator. Section 3 describes the models, the algorithm and the sampled quantities

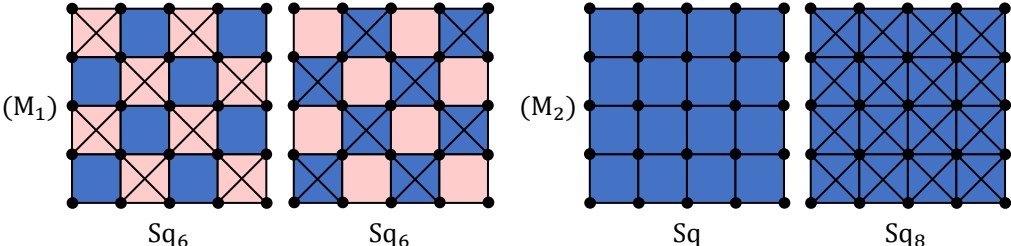

Figure 1: Two matching pairs constructed from the square lattice. For the left two figures ($M_1$), half of the elementary faces are chosen, and diagonals are added either to faces in the chosen set (left) or to those in its complement (right). The generated pair of matching lattices are isomorphic and are denoted $Sq_6$. For the right two figures ($M_2$), none of the squared faces is chosen, and the matching pair corresponds to the original square lattice (Sq) and a square lattice with both nearest- and next-nearest neighboring interactions ($Sq_8$).

from which we estimate the exponents. Section 4 presents the MC results for the one-point function of the NP operator, $W_{\mathrm{MNP}}(k)$, and the determination of its exponent $X_{\mathrm{MNP}}(k)$. Section 5 derives the universal scaling of the probability distribution of the MNP number, $\mathbb{P}_\ell$, on the basis of the scaling behavior of its generating function $W_{\mathrm{MNP}}(k)$, and then presents the MC results confirming these predictions. A brief discussion of our results is given in Sec. 6.

## 2 Exponent relations

In this section we shall demonstrate that $X_{\mathrm{PA}}(N) = X_{\mathrm{WM}}(N)$, the equality between the polychromatic $N$-arm exponent and the $N$-arm watermelon exponent. We will also relate the NP exponents to the NL exponents. Both arguments make use of a crucial property of site percolation on self-matching lattices: at the percolation threshold $p_c = 1/2$, occupied and empty sites play symmetric roles, and the color-inversion operation (occupied $\leftrightarrow$ empty) changes a critical configuration into another critical one.

### 2.1 Matching lattices

The concept of matching lattices plays an important role in percolation theory [3,5]. It is also an essential ingredient in the study of the NP operators: in the calculation of their exponents, in the proof [34] of the identity $W_{\mathrm{MNP}}(2) = 1$ for planar triangulation graphs of any size and shape, and in the algorithm for evaluating the nested-path number $\ell$.

We now briefly recall how to construct a pair of matching lattices. Given a planar graph $\mathcal{L}_0$, one selects an arbitrary set of elementary faces, and then generates a pair of graphs by adding any missing diagonal edges to each face in the chosen set (respectively to each face in the complementary set). In other (more graph theoretical) words, we replace each chosen face by the corresponding clique. The generated pair of graphs, denoted $\mathcal{L}$ and $\mathcal{L}^*$, has the same vertex set as the original one $\mathcal{L}_0$, and are called a *matching pair*.

It can be shown [5] that the site percolation thresholds for a matching pair of (regular and infinite) lattices satisfy $p_c + p_c^* = 1$. In particular, if the pair of matching graphs are isomorphic, $\mathcal{L} \cong \mathcal{L}^*$ the site-percolation threshold is $p_c = 1/2$. A lattice for which all faces are triangles already has all diagonals, so that $\mathcal{L} = \mathcal{L}^*$ for any choice of faces: such a lattice is called self-matching. Any planar triangulation graph, such as the triangular or the Union-Jack lattice, is self-matching and thus has $p_c = 1/2$.

Figure 1 shows two pairs of matching lattices constructed from the square lattice. In the

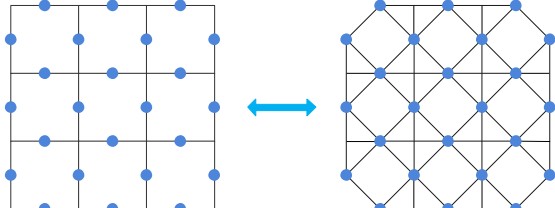

Figure 2: Illustration of the bond-to-site transformation. Each edge in the bond-percolation problem is transformed into a vertex (blue dot) in the site-percolation problem, and two sites are taken to be neighboring if the corresponding bonds are adjacent. This example maps bond percolation on the self-dual lattice (BSq) onto site percolation on the self-matching lattice ($Sq_6$).

right two figures, none of the elementary faces is chosen, and the matching pair consists of the original square lattice (Sq) and the square lattice with additional next-nearest neighbor interactions ($Sq_8$). Thus, the respective site-percolation thresholds satisfy $p_c(SSq) + p_c(SSq_8) = 1$. In the left two figures, the chosen set contains half of the square faces (shown in pink) in a checkerboard fashion, and the generated matching pair of lattices both have coordination number $z = 6$ and are isomorphic (they differ only by a rotation); we denote them as $Sq_6$. Moreover, by the bond-to-site transformation (defined in Fig. 2), it can be shown that site percolation $SSq_6$ is equivalent to bond percolation on the square lattice (BSq).

The construction of matching lattices for finite graphs is analogous to that for infinite graphs, except that special treatment is needed at boundaries. But the boundary effect is expected to play a vanishing role for percolation thresholds and bulk properties of systems.

## 2.2 Polychromatic $N$-arm exponent

Consider the configurations contributing to the one-point function of the polychromatic $N$-arm operator placed in the center of a domain. These configurations by definition support $N$ radial paths. At least one of these paths is on a primal cluster, and at least one path is on a dual cluster. If the arms strictly alternate between primal and dual clusters, it is clear that each pair of adjacent arms is separated by a radial cluster boundary. In this case also $N$ cluster boundaries connect the center to the rim. Conversely, the existence of $N$ radial cluster boundaries implies the existence of (at least) $N$ radial paths between them. When the disc is large enough, we may neglect the probability of having more than $N$ paths, because the exponents $X_{WM}(N)$ form a strictly monotonic sequence (i.e., the probability of having more paths decays algebraically faster). As a consequence, for the case in which the polychromatic arms alternate in color, $X_{PA}(N) = X_{WM}(N)$. Although it is conceivable, in principle, that the exponent $X_{PA}(N)$ depends on the precise (cyclic) sequence of primal and dual arms, we will argue below that the exponent $X_{PA}(N) = X_{WM}(N)$ irrespective of this sequence, by elaborating on the ideas of [20].

We start by introducing a construction to allow for the $N$-arm WM operator with odd $N$, where primal and dual clusters exchange roles for an observer passing around the insertion point. Thus for the WM operator inserted in the center of a domain, we must include the possibility of anticyclic symmetry. We note that this is only possible in a self-matching lattice model. We allow anticyclic symmetry by introducing a radial chain of sites which from one side of the chain are seen as primal, and from the other as dual or vice versa. We call such chain an *inversion chain*. It is illustrated in Fig. 3b, where the three cluster boundaries are shown in bold white. An inversion chain can be moved around (while keeping its end points fixed) without affecting the position of the cluster boundaries. Therefore two radial inversion

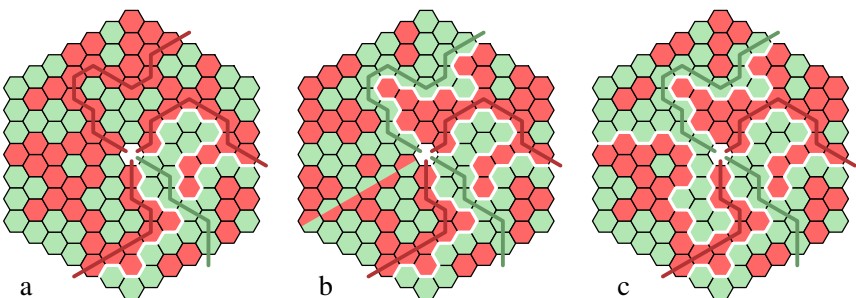

Figure 3: Configurations contributing to the one-point function of the polychromatic 4-arm operator. A bijective transformation described in the text, a↔b and b↔c, adds or removes a radial cluster boundary, here indicated in bold white.

chains can be moved to coincide, thus annihilating each other. Moreover we do not consider the locus of the inversion chain as being defined by the configuration.

Let us now consider a configuration contributing to the polychromatic $N$-arm one-point function in a self-matching lattice model for site percolation, with the operator inserted in the center of a domain with the topology of a disc. There must be at least one radial cluster boundary separating a primal path and a dual path. An example is shown in Fig. 3a, contributing to the one-point function of the polychromatic ($N$=4)-arm operator, as it has four radial paths from the center to the rim, three red (primal) and one green (dual), and two radial cluster boundaries. We choose a radial cluster boundary, in the example, the one ending on the right-most side of the hexagonal domain. From this cluster boundary in the positive (anti-clockwise) direction, we consider the adjacent path, primal or dual. Unlike a cluster boundary, a path is not uniquely defined by the configuration. We choose the *closest path*, i.e. the path as close as possible to the cluster boundary: all its elements touch the cluster boundary. Then, we switch, from occupied to unoccupied or vice versa, all the elements that lie in the positive direction from this path (not including the path itself) until an inversion chain that is either created or annihilated in this flipping operation. In the transformation shown in Fig. 3 a→b, an inversion chain is created running straight from the center to the lower-left corner of the domain. It is immaterial where the inversion chain is positioned. In the case that an inversion chain already exists and intersects the closest path, it may first be moved to a position without such overlap to avoid ambiguity.

If the second path in the positive direction from the chosen domain wall has the same color as the first path, a new radial domain wall is created by this flipping operation. If they are different, a domain wall disappears between the two paths. Examples of these two cases are the transformations in Fig. 3 from a to b and from b to c respectively. Note that the flipping operation by its definition is bijective, since the defining objects: a given radial cluster boundary, and its closest radial path in the positive direction remain unchanged in the operation.

By this flipping operation, any configuration contributing to the one-point function of the $N$-arm PA operator with the coloring of the arms in some arbitrary order, can be turned bijectively into a configuration of the corresponding one-point function with primal and dual arms strictly alternating. As a consequence $X_{\text{PA}}(N) = X_{\text{WM}}(N)$ irrespective of the order in which the primal and dual paths follow each other around the PA operator, provided there is at least one radial cluster boundary. We note that this argument was presumably implicit in Ref. [20], but the details were only sketched very briefly there.

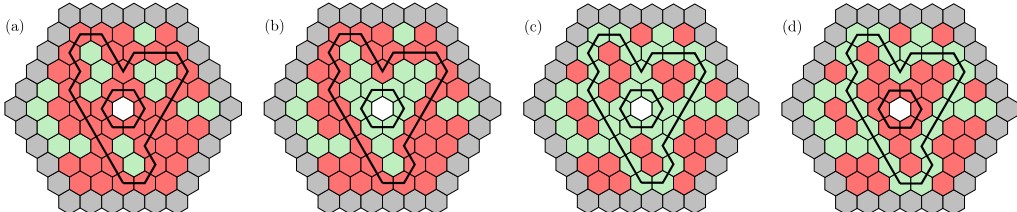

Figure 4: Examples of a set of site configurations on the triangular lattice with $\ell = 2$ nested closed paths. The central site is neutral (white), the occupied (empty) sites are represented as red (green) sites, and the sites on the fixed boundary are marked gray. The map $P_1$, associated with the first NP, leads to (a) $\leftrightarrow$ (b) and (c) $\leftrightarrow$ (d), while $P_2$ leads to (a) $\leftrightarrow$ (c) and (b) $\leftrightarrow$ (d). For a given statistical weight $k$, all the four configurations contribute to the one-point NL function $W_{\mathrm{NL}}(k)$, with a total amount $1 + k + k + k^2 = (k+1)^2$. By comparison, only (a) contributes to the one-point NP function $W_{\mathrm{NP}}(k)$ with an amount $k^2$.

## 2.3 Nested-path exponent

We next show how to calculate the NP exponents by rigorousy relating the one-point functions of the NP operators to the one-point function of the NL operator. A so-called color-inverting technique, similar to the one used above in the argument for establishing the identity $X_{\mathrm{PA}}(N) = X_{\mathrm{WM}}(N)$, is applied to site percolation on a self-matching lattice. By universality we assume the result to be true also for bond percolation, and for other regular 2D lattices.

We take a domain with a fixed boundary condition, i.e. the sites on the boundary of the domain are all occupied (or all unoccupied). We note however, that the boundary condition only affects $W_{\mathrm{NL}}$, not the $W_{\mathrm{NP}}$ themselves. The fixed boundary condition ensures that all cluster boundaries are closed loops.

All percolation configurations contribute to $W_{\mathrm{NL}}(k)$, with a weight $k$ for each nested loop. We first focus on the complete set of nested paths, not necessarily all of the same color. We make the NPs unique by choosing each one closest to the interior NP, starting with the inner-most NP closest to the center; the exact algorithm for doing so is provided in Ref. [34] and discussed further in Sec. 3. We introduce the transformations $P_j$, that flip all the sites of the $j$-th path (counted from the center) and all the sites interior to it. Thus a configuration with $\ell$ polychromatic NPs, is a member of a set of $2^\ell$ configurations, generated by (all subsets of) the $P_j$ acting on it. An example is given in Fig. 4, where there is a total number of $2^\ell = 4$ configurations ($\ell = 2$). The ensemble of all configurations is the disjoint union of these sets. Whenever two consecutive NPs or the outermost NP and the boundary are colored differently they are separated by an NL. This mechanism accounts for all possible NLs. The total contribution of the set of configurations to $W_{\mathrm{NL}}(k)$ is $(k + 1)^\ell$ as each $P_j$ increases or decreases the number of NLs by one. Of the set of configurations only one contributes to $W_{\mathrm{MNP}}(k)$, namely the one in which each of the nested paths is occupied. By setting the weight of the MNPs to $(k + 1)$, the two one-point operators are equal, $W_{\mathrm{MNP}}(k + 1) = W_{\mathrm{NL}}(k)$, or for the exponents

$$X_{\mathrm{MNP}}(k) = X_{\mathrm{NL}}(k - 1) \tag{5}$$

To the one-point function of the PNP operator, $W_{\mathrm{PNP}}$, all $2^\ell$ configurations contribute equally, as they are all equiprobable and have $\ell$ PNPs. The total contribution thus agrees with that of $W_{\mathrm{NL}}$ if the PNPs have weight $(k + 1)/2$, leading to the exponent relation

$$X_{\mathrm{PNP}}(k) = X_{\mathrm{NL}}(2k - 1) \tag{6}$$

In view of the expression for $X_{\mathrm{NL}}$ (2), this leads to the expressions (3) and (4) respectively.

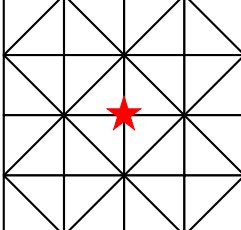 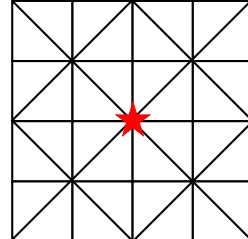

Figure 5: Union Jack lattice with the center site (denoted by the red star) on different sublattices ($UJ_4$ and $UJ_8$). The coordination number of the center site is 4 for the left and 8 for the right.

This simple relation is also valid for a domain with a free boundary condition, with a center which itself is occupied. In this case the same argument holds, with an alternative definition of the $P_j$: flipping the $j$-th NP and all the sites *exterior* of it.

# 3   Model, algorithm and sampled quantities

Apart from bond percolation on the square lattice (BSq), we also consider site percolation on the triangular lattice (STr), and on four other lattices, which are the Union-Jack (UJ) lattice with the center site respectively on each of the two sublattices ($SUJ_4$ and $SUJ_8$), and the square lattice with only nearest- (SSq) and with both nearest- and next-nearest neighbor interactions ($SSq_8$), respectively. The subscript of SUJ specifies the coordination number $z$ for the sublattice with the center site, as illustrated in Fig. 5. In the thermodynamic limit ($L \to \infty$), SSq and $SSq_8$ are matching to each other, and all the others are self-matching (see Sec. 2.1).

Since only the MNP operator will be considered from now on, we avoid heavy notation using the symbols $\mathcal{W}_k$ or $\mathcal{W}_k(L)$ for the MNP correlator and $X$ for the MNP exponent, instead of $W_{\mathrm{MNP}}(k)$ and $X_{\mathrm{MNP}}$. Also the label NP will typically refer to monochromatic nested paths, unless explicitly specified otherwise.

## 3.1   Algorithm

In this work, percolation is studied on a domain with the topology of a disc, with free boundary conditions. The domain shape is chosen to be hexagonal for the triangular lattice and square for the others. The scale $L$ is the length of the corner-to-corner diagonal for the former, and the side length for the latter. Figure 4(a) shows an example configuration for STr with $L = 9$. The central site is neutral and the other sites are occupied with the critical probability $p_c = 1/2$.

Meanwhile, we consider only the *central* cluster that contains the central site or bond, and use $\mathcal{R} = 1$ to specify the percolating event that the central cluster reaches the boundary, and otherwise we set $\mathcal{R} = 0$. For the $\mathcal{R} = 1$ case, we calculate the maximum number $\ell$ of independent closed paths in the central cluster that surrounds the center. We stress that, while the number $\ell$ of nested paths is well defined, their locations might not be unique. Thus, the method used to evaluate $\ell$ need not specify the location of the paths uniquely, but it must guarantee that it is not possible to find a larger number of nested paths in the given configuration.

By carefully examining Fig. 4(a) for the triangular-lattice site percolation, we observe that a unique innermost nested path can be identified. By growing the *matching* cluster of empty sites starting from the center and terminating when the cluster cannot be grown any further, one obtains the first nested path as the outer boundary of the matching cluster. Similarly, the

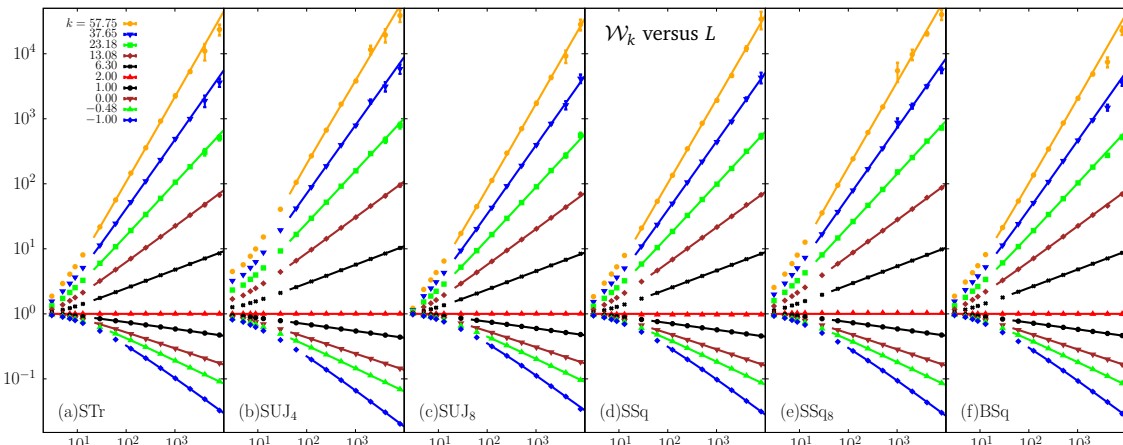

Figure 6: Log-log plot of one-point MNP correlator $\mathcal{W}_k$ versus linear size $L$, for (a) STr, (b) SUJ$_4$, (c) SUJ$_8$, (d) SSq, (e) SSq$_8$ and (f) BSq. The lines represent fits to Eq. (7) and strongly indicate the algebraic dependence of $\mathcal{W}_k$ on $L$. Moreover, the striking similarity exhibited by the six models clearly supports the universality of $\mathcal{W}_k$.

second nested path can be regarded as the outer boundary of the matching clusters which are linked together by the first nested path. In other words, by growing the matching clusters from the first closed path, one can locate the second nested path as the chain of occupied sites that stops the matching-cluster growth. The procedure is repeated until the growth of matching clusters reaches the open boundary of the domain. By this method, we obtain a specific and complete set of independent nested paths, and in particular the number $\ell$ of nested paths.

The procedure works for site percolation on any self-matching lattice $\mathcal{L} = \mathcal{L}^*$. For a non-self-matching lattice $\mathcal{L}$, the matching clusters of empty sites must be defined on the corresponding matching lattice $\mathcal{L}^*$. For instance, to evaluate $\ell$ for SSq, matching clusters are grown on SSq$_8$, and vice versa. The procedure is similar for bond percolation, where the nested path is now defined as the chain of occupied bonds that stops the growth of *dual* clusters that live on the dual lattice.

## 3.2 Sampled quantities

For each configuration at criticality, we record the percolation indicator $\mathcal{R}$ and, if $\mathcal{R} = 1$, evaluate the MNP number $\ell$. On this basis, we calculate and study:

1. The probability distribution $\mathbb{P}_\ell(L)$ of having $\ell$ closed, monochromatic nested paths in the percolating cluster ($\mathcal{R} = 1$), each surrounding the center. By definition, $\sum_{\ell \geq 0} \mathbb{P}_\ell = \langle \mathcal{R} \rangle$.

2. The one-point MNP correlator $\mathcal{W}_k \equiv \langle \mathcal{R} \cdot k^\ell \rangle \equiv \sum_{\ell=0} k^\ell \mathbb{P}_\ell$, where the NP fugacity $k \in \mathbb{R}$ (by convention, $0^0 = 1$ for $k = 0$). Notice that, once the $\mathbb{P}_\ell$ have been computed in the simulations, the $\mathcal{W}_k$ for any $k$ can be readily calculated afterwards. At criticality, it has been observed for STr and BSq [34] that $\mathcal{W}_k$ depends on $L$ as $L^{-X_{\text{MNP}}}$.

3. The conditional NP number $N \equiv \langle \mathcal{R} \cdot \ell \rangle / \langle \mathcal{R} \rangle$. This is the average number of independent nested paths conditioned by the existence of a percolating cluster.

4. The probability ratio $\Gamma_\ell = (\ell! \, \mathbb{P}_\ell / \mathbb{P}_0)^{1/\ell}$ for $\ell \geq 1$.

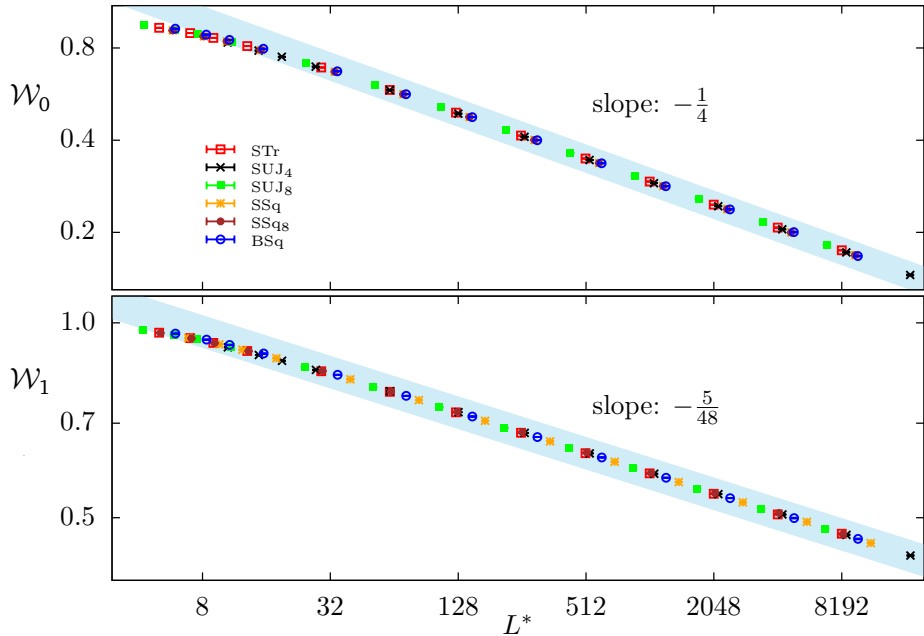

Figure 7: Log-log plot of $\mathcal{W}_0$ (top) and $\mathcal{W}_1$ (bottom) versus rescaled size $L^* = aL$, where $a$ is a model-dependent constant (we fix $a = 1$ for STr). By fine-tuning the value of $a$, the MC data for all the six models collapse onto an asymptotically straight line, with slope $-1/4$ for $\mathcal{W}_0$ and $-5/48$ for $\mathcal{W}_1$. This strongly supports the universality of $\mathcal{W}_k$, at least for $k = 0$ and $1$.

## 4 Numerical results for the one-point function

Simulations were carried out at the percolation threshold, which is $p_c = 1/2$ for BSq, STr, SUJ$_4$ and SUJ$_8$. For SSq, albeit the exact value of $p_c$ is still unknown, it has been determined with a high precision as $p_c = 0.592\,746\,050\,792\,10(2)$ [35–37]; for SSq$_8$, the self-matching argument gives $p_c(\text{SSq}_8) = 1 - p_c(\text{SSq})$. The linear system size $L$ was taken in the range $3 \leq L \leq 8189$. For each system, and for each $L$, the number of samples is at least $5 \times 10^9$ for $L \leq 100$, $2 \times 10^8$ for $100 < L \leq 1000$, $2 \times 10^7$ for $100 < L \leq 4000$, and $5 \times 10^6$ for $L > 4000$.

### 4.1 Scaling and universality of $\mathcal{W}_k$

For the one-point NP correlation functions $\mathcal{W}_k(L)$, Fig. 6 displays the MC data versus the linear size $L$ for all the six percolation models considered in this work. For $k < 0$, the contributions to $\mathcal{W}_k$ from even and odd values of $\ell$ partly compensate, and thus the relative error margin becomes larger as $k$ decreases. As a consequence, for large negative $k$ it is difficult to obtain meaningful data (with small relative error bars for $\mathcal{W}_k$). Further, finite-size corrections for small $L$ become more pronounced as $k$ decreases.

As will be shown in Sec. 5, it is observed that, for any given size $L$, the probability distribution $\mathbb{P}_\ell$ would vanish super-exponentially fast as the NP number $\ell$ increases. This means that, given any finite $k$ and $L$, the series $k^\ell \mathbb{P}_\ell$ is always convergent and thus the NP correlator $\mathcal{W}_k \equiv \sum k^\ell \mathbb{P}_\ell$ is always well defined. Nevertheless, as $k$ increases, the contribution from large $\ell$ becomes more important. In practice, the MC method is not well suited for sampling a large number with a small probability and can in principle introduce bias in the estimate of error bars if the number of samples is not sufficiently big. Therefore, with our current simulations, we cannot calculate $\mathcal{W}_k$ for very large $k$, and thus restrain to values $k < 60$.

The approximate linearity of the log-log plot in Fig. 6 demonstrates the expected power-law

| | | $L_\mathrm{m}$ | $\chi^2$/DF | $X$ | $c_0$ | $c_1$ | $c_2$ |
|---|---|---|---|---|---|---|---|
| $\mathcal{W}_1$ | STr | 29 | 0.98 | 0.1043(2) | 1.206(1) | -0.22(4) | -1.3(6) |
| | | 61 | 1.22 | 0.1043(4) | 1.205(3) | -0.2(2) | -2(6) |
| | $SUJ_4$ | 29 | 0.48 | 0.1043(1) | 1.120(1) | -0.17(3) | -0.3(4) |
| | | 61 | 0.58 | 0.1043(3) | 1.120(2) | -0.2(1) | -1(4) |
| | $SUJ_8$ | 29 | 1.28 | 0.1042(2) | 1.227(1) | -0.24(4) | -2.1(6) |
| | | 61 | 0.59 | 0.1040(3) | 1.225(2) | -0.1(1) | -6(5) |
| | SSq | 29 | 1.00 | 0.1042(2) | 1.168(1) | -0.14(4) | -1.3(6) |
| | | 61 | 1.10 | 0.1043(4) | 1.169(3) | -0.2(2) | 1(5) |
| | $SSq_8$ | 29 | 0.94 | 0.1042(2) | 1.207(1) | -0.24(4) | -1.6(6) |
| | | 61 | 1.15 | 0.1042(3) | 1.207(3) | -0.3(2) | -1(5) |
| | BSq | 29 | 0.97 | 0.1042(2) | 1.186(1) | -0.12(4) | -1.2(6) |
| | | 61 | 0.90 | 0.1044(3) | 1.188(3) | -0.2(1) | 2(5) |
| $\mathcal{W}_0$ | STr | 29 | 0.78 | 0.2500(3) | 1.658(3) | -1.64(8) | 1(1) |
| | | 61 | 0.96 | 0.2500(4) | 1.658(7) | -1.6(4) | 0(8) |
| | $SUJ_4$ | 29 | 0.49 | 0.2503(2) | 1.379(2) | -0.76(6) | 0.1(8) |
| | | 61 | 0.37 | 0.2500(4) | 1.377(4) | -0.6(2) | -5(6) |
| | $SUJ_8$ | 29 | 0.92 | 0.2500(3) | 1.731(3) | -2.03(8) | 2(1) |
| | | 61 | 0.26 | 0.2495(3) | 1.726(3) | -1.7(2) | -9(6) |
| | SSq | 29 | 0.31 | 0.2501(2) | 1.604(2) | -1.51(5) | 1.4(8) |
| | | 61 | 0.28 | 0.2503(3) | 1.606(3) | -1.6(2) | 6(6) |
| | $SSq_8$ | 29 | 1.15 | 0.2499(3) | 1.602(3) | -1.48(9) | 1(1) |
| | | 61 | 1.10 | 0.2502(6) | 1.606(6) | -1.7(4) | 8(8) |
| | BSq | 29 | 1.08 | 0.2500(3) | 1.600(3) | -1.2(1) | 1(1) |
| | | 61 | 0.91 | 0.2503(6) | 1.604(6) | -1.5(4) | 9(8) |

Table 1: Fitting results for $\mathcal{W}_1$ and $\mathcal{W}_0$ by Eq. (7) with correction exponent $\omega = 1$. The exponents $X(1)$ and $X(0)$ are well consistent with the exact value $5/48 \approx 0.10417\cdots$ and $1/4$.

scaling $\mathcal{W}_k(L) \sim L^{-X}$ for a broad range of $k \geq -1$. Moreover, the striking similarity exhibited by the different percolation models clearly indicates that the scaling of the NP correlations does not depend on microscopic details, and is thus universal.

## 4.2 The $k = 0, 1$ cases

As discussed earlier, $\mathcal{W}_1$ reduces to the percolating probability $\langle \mathcal{R} \rangle$, which is known to decay as $\mathcal{W}_1 \sim L^{-X_\mathrm{F}} = L^{-5/48}$, and $\mathcal{W}_0$ corresponds to the polychromatic two-arm correlation, with exponent $X(2) = X_\mathrm{WM}(2) = 1/4$. The universality of $\mathcal{W}_1$ and $\mathcal{W}_0$ is further illustrated in Fig. 7. With a rescaled linear size $L^* = aL$ where $a$ is a model-dependent constant of order unity, the MC data of $\mathcal{W}_1$ for different systems collapse nicely onto an asymptotically straight line, upon fine-tuning $a$. The same holds true for $\mathcal{W}_0$.

We fit the $\mathcal{W}_k$ data, according to the least-squares criterion, to the form

$$\mathcal{W}_k = L^{-X}(c_0 + c_1 L^{-\omega} + c_2 L^{-2\omega}), \tag{7}$$

where the terms with $c_1$ and $c_2$ account for finite-size corrections. We impose a lower cutoff $L \geq L_\mathrm{m}$ on the data points admitted in the fits, and systematically study the effect on the residual $\chi^2$-value upon increasing $L_\mathrm{m}$. For percolation systems with free boundary conditions, one generally expects the correction exponent $\omega = 1$. With $\omega = 1$, the fitting results are shown in Table 1 for $\mathcal{W}_1$ and $\mathcal{W}_0$. The estimates of $X(1)$ and $X(0)$ agree excellently with the exact values, which are $5/48$ and $1/4$ respectively. The fits with $\omega$ being a free parameter give consistent results and $\omega \approx 1$.

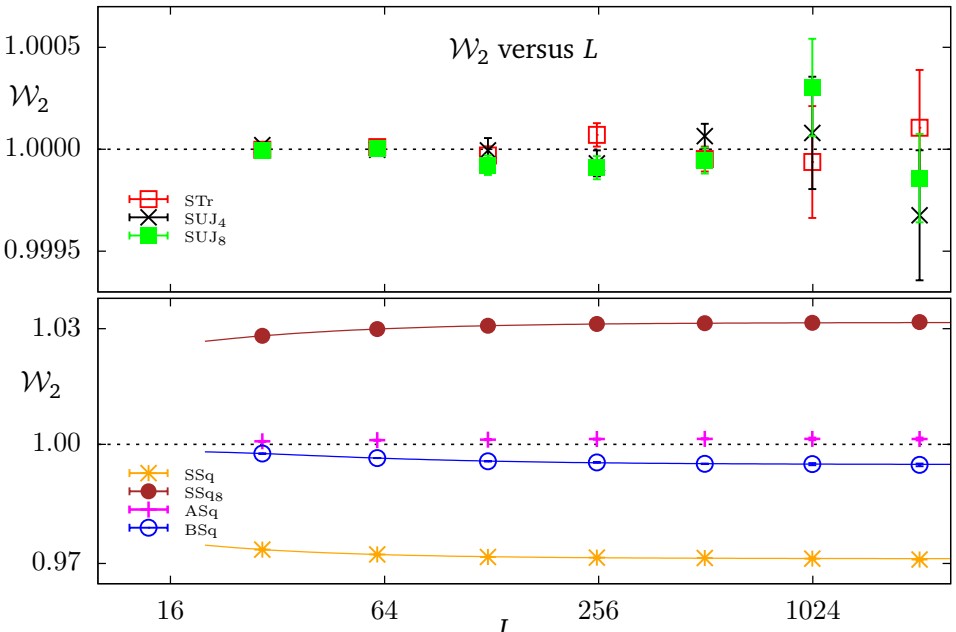

Figure 8: Plot of $\mathcal{W}_2$ versus the linear size $L$ for STr, $\mathrm{SUJ}_4$ and $\mathrm{SUJ}_8$ (top) and for BSq, SSq and $\mathrm{SSq}_8$ (bottom). Also shown are the averaged data for SSq and $\mathrm{SSq}_8$, denoted by "ASq". For triangulation lattices (top), the value of $\mathcal{W}_2(L)$ is exactly 1 for any $L$, while for other lattices, $\mathcal{W}_2(L \to \infty)$ converges to some constant slightly away from 1. The curves, obtained from the least-squares fits, are guides to the eye.

|  | $L_{\mathrm{m}}$ | $\chi^2/\mathrm{DF}$ | $c_0$ | $c_1$ | $c_2$ |
|---|---|---|---|---|---|
| SSq | 29 | 0.73 | 0.9712(1) | 0.063(8) | 0.1(2) |
|  | 61 | 0.50 | 0.9713(2) | 0.04(2) | 1(2) |
| $\mathrm{SSq}_8$ | 29 | 0.14 | 1.03165(6) | -0.110(4) | 0.22(8) |
|  | 61 | 0.13 | 1.0317(1) | -0.12(2) | 0.7(8) |
| BSq | 29 | 0.99 | 0.9948(1) | 0.12(1) | -1.1(2) |
|  | 61 | 1.24 | 0.9949(2) | 0.11(5) | -1(2) |

Table 2: Fitting results of $\mathcal{W}_2$ for SSq, $\mathrm{SSq}_8$ and BSq, by Eq. (7) with $X(2) = 0$. The asymptotic values $c_0 \equiv \mathcal{W}_2(L \to \infty)$ and the averaged value 1.0015(2) for SSq and $\mathrm{SSq}_8$ are slightly but clearly different from 1.

### 4.3 The $k = 2$ case

By definition, $\mathcal{W}_k$ is an increasing function of $k$. It is thus expected that a special value $k_s$ exists such that, as $L$ increases, $\mathcal{W}_k(L)$ decays for $k < k_s$, diverges for $k > k_s$, and converges to some constant for $k = k_s$. From Ref. [34], it is known that $k_s = 2$, and this is well confirmed by Fig. 6.

The MC data of $\mathcal{W}_2(L)$, plotted in Fig. 8, give further strong evidence for $k_s = 2$. For the three site-percolation systems on the triangulation lattices (STr, $\mathrm{SUJ}_4$ and $\mathrm{SUJ}_8$), the MC data, with a precision of the order $\mathcal{O}(10^{-6})$ for some sizes, suggest that $\mathcal{W}_2(L) = 1$ for any $L$. This observation is further supported by the exact enumeration results, which are obtained for $L = 3, 5, 7$ for STr and $L = 3, 5$ for $\mathrm{SUJ}_4$ and $\mathrm{SUJ}_8$. For the three other percolation models (BSq, SSq and $\mathrm{SSq}_8$), the $\mathcal{W}_2(L)$ value also converges to some constant, which is slightly but clearly different from 1. We fit the $\mathcal{W}_2(L)$ data to Eq. (7), according to the least-squares criterion, by fixing $X = 0$, and the results are given in Table 2.

Motivated by the observation that $\mathcal{W}_2(L)$, as obtained from either MC simulations or exact enumerations, is consistent with 1 for STr for any $L$, the authors of Ref. [34] proved that

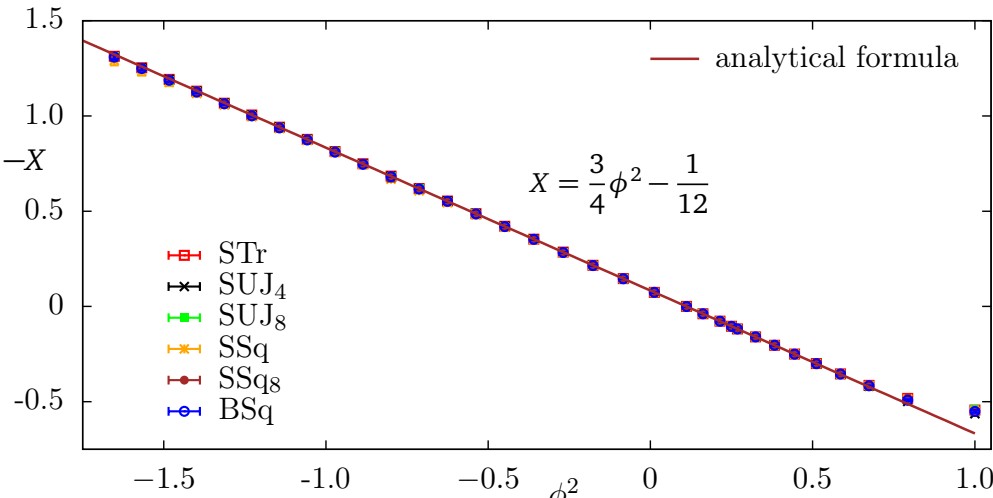

Figure 9: The MNP exponent $-X \equiv -X_{\mathrm{MNP}}$ versus parameter $\phi^2$. Estimates of $X$ for all the six percolation systems agree very well with Eq. (3) for a broad range of $k$. The analytical formula is represented by the brown line.

| $k$ | 23.18 | 15.26 | 5.02 | -0.48 | -0.69 | -1 |
|------|---------|---------|-----------|----------|----------|----------|
| BSq | -0.810(2) | -0.617(1) | -0.215(2) | 0.355(1) | 0.416(2) | 0.551(6) |
| STr | -0.813(3) | -0.619(1) | -0.2163(6) | 0.354(1) | 0.414(2) | 0.544(6) |
| SUJ$_4$ | -0.813(4) | -0.619(2) | -0.2167(2) | 0.356(1) | 0.419(2) | 0.564(9) |
| SUJ$_8$ | -0.812(3) | -0.618(1) | -0.2165(2) | 0.355(2) | 0.416(3) | 0.543(5) |
| SSq | -0.808(6) | -0.607(8) | -0.215(1) | 0.355(1) | 0.416(1) | 0.549(6) |
| SSq$_8$ | -0.812(3) | -0.61(1) | -0.216(2) | 0.354(1) | 0.415(2) | 0.548(7) |
| Theory | -0.8123 | -0.6180 | -0.2163 | 0.3558 | 0.4215 | 0.6667 |

Table 3: Some results for the fit of the NP exponent $X(k)$. The last row contains the theoretical prediction of Eq. (3). The fitting results $X(-1) = 0.548(7)$ is smaller than the predicted value 2/3 by about fifteen error bars. This indicates that the fitting formula (7) is not sufficient to describe the $\mathcal{W}_{-1}(L)$ data.

460   indeed $\mathcal{W}_2(L) = 1$ for site percolation on regular or irregular planar triangulation graphs, of
461   any shape and position of the centering site. This proof eventually led to the more general
462   proof given in Sec. 2.3.
463      A natural question arises for bond percolation on the self-dual square lattice (BSq), which
464   also has $p_c = 1/2$. Further, as illustrated in Fig. 2, it can be regarded as site percolation on the
465   lattice Sq6, a lattice isomorphic to its matching lattice. With the same squared shape as in [34],
466   it is found from Fig. 8 and Table 2 that $\mathcal{W}_2(L)$ depends non-trivially on $L$, and the asymptotic
467   value $\mathcal{W}_2(L \to \infty)$ is different from 1. We have studied BSq with other domain shapes,
468   arriving at the same observations. Further, for different domain shapes, the asymptotic values
469   of $\mathcal{W}_2(L \to \infty)$ are different. The appendix in Ref. [34] provides some further analytical
470   discussions on $\mathcal{W}_2(L)$ for BSq.
471      Figure 8 shows that $\mathcal{W}_2(L) < 1$ for SSq and $\mathcal{W}_2(L) > 1$ for SSq$_8$. Since these lattices
472   are mutually matching in the $L \to \infty$ limit, we calculate the average values of their $\mathcal{W}_2(L)$,
473   denoted as "ASq" in Fig. 8. This value is very close to, but still different from 1.
474      In short, despite the fact that $\mathcal{W}_2(L) = 1$ for self-matching triangulation graphs, the asymp-
475   totic value $\mathcal{W}_2(L \to \infty)$ is in general non-universal and depends on lattice types, domain
476   shapes, and the location of the center.

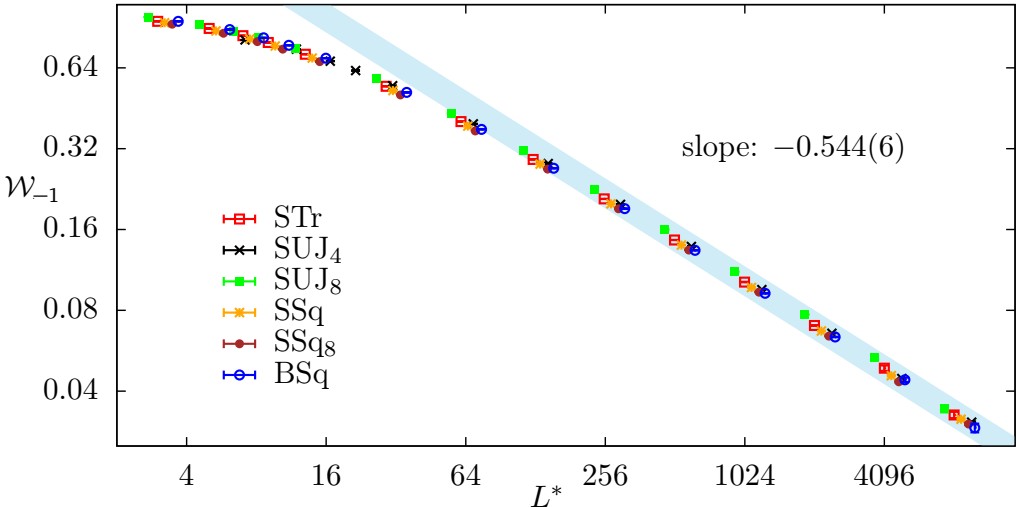

Figure 10: Log-log plot of $\mathcal{W}_{-1}$ versus rescaled size $L^* = aL$, where $a$ is a model-dependent constant ($a = 1$ for STr). The formula (3) predicts the slope to be $2/3$, while the numerical estimate is $0.544(6)$ for STr. The huge difference indicates that Eq. (7) is not sufficient to describe the $\mathcal{W}_{-1}$ data.

## 4.4 Nested-path exponent

In Sec. 2.3, we have derived the analytical formula (3) for the NP exponent $X(k)$, where $k$ is parameterized as $k = 1 + 2\cos(\pi\phi)$. For $-1 \leq k \leq 3$, $\phi$ has a real solution in the range $0 \leq \phi \leq 1$, and the known exact values are $X(0) = 1/4$, $X(1) = 5/48$ and $X(2) = 0$. For $k > 3$, $\phi$ becomes purely imaginary, and, letting $\phi = i\alpha$ yields $k = 1 + 2\cosh(\pi\alpha)$. For $k < -1$, $\phi^2$ is not real, and, mostly probably, one has no longer the power-law scaling behavior as $\mathcal{W}_k(L) \sim L^{-X}$.

Figure 6 shows the numerical results of $\mathcal{W}_k$ versus $L$, for a broad range of $k$. The expected power-law scaling is clearly observed, though strong finite-size corrections exist for $k \in [-1, -0.5]$. Furthermore, the $\mathcal{W}_k(L)$ data are well described by Eq. (7) for most $k$ with correction exponent $\omega = 1$ for reasonable values of $L_m$. The details of the fits are described in the appendix, and the results of $X(k)$, from the six percolation models, are plotted versus $\phi^2$ in Fig. 9. For convenience of comparison, the results for some values of $k$ are also listed in Table 3. It is shown that the estimated values of $X$ for different percolation systems are consistent with each other within the error bars. This demonstrates the universality of the critical behavior associated with nested paths. Moreover, except the last three data points that are for $k \simeq -0.69, -0.88$ and $k = -1$, the estimates of $X(k)$ are in good agreement with the prediction by Eq. (3). Also for $k \simeq -0.69$, the agreement between the numerical and theoretical results is acceptable, to within twice the quoted error bars.

It is interesting to note that, if our numerical estimates of $X(k)$ were compared to the previously conjectured formula in [34], the agreement would also look good, even for small values $k \in [-1, 0.5]$. For instance, for the $k = -1$ case, the fit by Eq. (7) yields $X(-1) \approx 0.544$, which nicely agrees with the conjectured value $13/24 \approx 0.542$. Further, as shown in Fig. 10, the $\mathcal{W}_{-1}$ data versus a rescaled size $L^* = aL$ in the log-log scale, collapse onto an asymptotically straight line for all the six percolation systems. Actually, for $k$ slightly smaller than $-1$, one can still obtain reasonably good fits by Eq. (7). This interesting fact is a warning that, without theoretical guidance, the fitting of ill-behaved numerical data may produce misleading results.

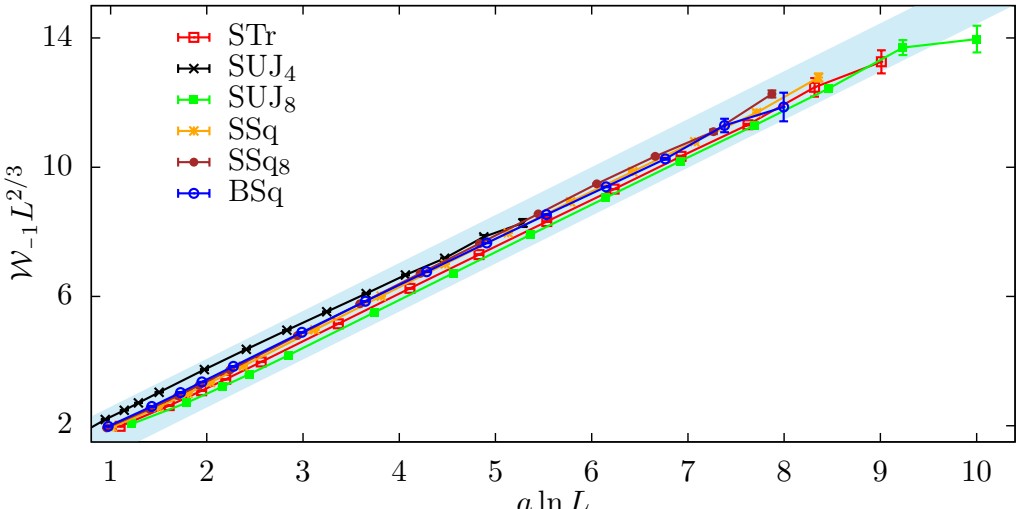

Figure 11: Illustration of the logarithmic correction by plotting $\mathcal{W}_{-1}L^{2/3}$ versus $a\ln L$, with $a$ a model-dependent constant (we set $a=1$ for STr). The approximate linearity for all systems supports $\mathcal{W}_{-1}\sim L^{-2/3}(\ln L)$.

### 4.5 The $k=-1$ case

We reexamine the finite-size scaling analysis for the $\mathcal{W}_{-1}(L)$ data by noticing the following general picture. As the statistical weight $k$ decreases, the one-point NP correlation function exhibits algebraic scaling behavior for $k > k_c = -1$, and then enter into a 'disordered' phase in which $\mathcal{W}_k(L)$ vanishes exponentially as $L$ increases. This behavior is reminiscent of that observed for a phase transition between a quasi-long-range ordered phase and a disordered phase, occuring in particular in the Berezinskii-Kosterlitz-Thouless (BKT) phase transition. In this analogy, the special value $k_c = -1$ acts as the BKT transition point. There is another interesting fact exhibited by the NP exponent as a function of $k$: from Eq. (3), one observes that the derivative of $X(k)$ with respect to $k$ diverges at $k_c = -1$. At the critical point $k_c$, one might expect that the power-law scaling behavior of the one-point correlation $\mathcal{W}_k(L)$ is modified by additive and multiplicative logarithmic corrections.

Since the numerical estimate $X_{\text{NP}}(-1) \approx 0.544$ is significantly smaller than the theoretical value $2/3$, we simply assume that a multiplicative logarithmic correction arises as $\mathcal{W}_{-1}(L) \sim L^{-2/3}(\ln L)$. Fig. 11 shows a plot of the $\mathcal{W}_{-1}L^{2/3}$ data versus $a\ln L$, with $a$ a model-dependent rescaling constant. It can be seen that the data for all the six percolation systems collapse resonably well onto an approximately straight line.

Furthermore, by assuming some analogy with the BKT phase transition and borrowing insights from the latter, we can try to make a finite-size scaling analysis for the $\mathcal{W}_k(L)$ data for some range $k < k_c$. According to the BKT theory, as the BKT transition point is approached from the disordered phase, the correlation length $\xi$ would diverge exponentially as $\xi \sim \exp(c/\sqrt{t})$, where $t$ represents the distance to the criticality and $c$ is a non-universal constant. For any physical observable $Q$, the finite-size scaling near criticality would behave as $Q(t,L) \sim L^Y \tilde{Q}(t\ln^2 L)$, where $Y$ is the corresponding exponent. Accordingly, in Fig. 12 we plot $\mathcal{W}_k L^{2/3}/\ln(L/L_1)$ versus $(k-k_c)\ln^2(L/L_0)$, where $L_0$ and $L_1$ are model-dependent constants. Indeed, the numerical data for different system sizes more or less collapse onto each other. Despite of its incomplete theoretical foundations, this analysis indicates that $k_c = -1$ seems to behave like a BKT transition point, and we conclude that logarithmic corrections are likely to exist at $k_c$.

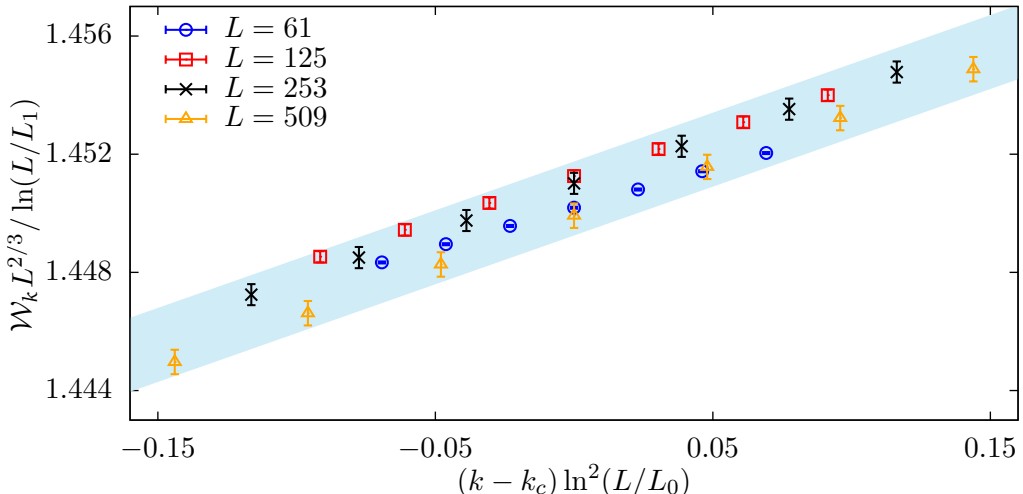

Figure 12: $\mathcal{W}_k L^{2/3} / \ln(L/L_1)$ versus $(k - k_c) \ln^2(L/L_0)$ for STr, where $L_0$ and $L_1$ are non-universal constants. The more-or-less collapse of numerical data for different sizes indicates that $k_c = -1$ seems to behave like a BKT transition point.

## 5 Probability distribution

We now consider the probability distribution $\mathbb{P}_\ell$ that the center cluster is percolating ($\mathcal{R} = 1$) and has $\ell$ independent closed nested paths (NPs) surrounding the center. The MC results for the six percolation systems are given in the appendix, and, as an example, the results for STr are shown in Fig. 13. It indicates that $\mathbb{P}_\ell$ vanishes super-exponentially fast as a function of $\ell$. Actually, the number of NPs detected in our current simulations is limited to $\ell \leq 5$, even for $L = 8189$. For $\ell = 0$, the algebraic decay, $\mathbb{P}_0 \sim L^{-1/4}$, is consistent with the approximately linear decrease on the logarithmic scale used in Fig. 13. For $\ell = 1$, Table 6 in the appendix tells that $\mathbb{P}_1(L)$ first increases with $L$ but then starts decreasing; actually, this can be seen by zooming in on Fig. 13 since the error bars are much smaller than the symbol size for the $\ell = 1$ data points. For $\ell \geq 2$, $\mathbb{P}_\ell(L)$ increases as a function of $L$ within the current range $253 \leq L \leq 8189$ of simulations, but the increasing speed seems to slow down. This makes us suspect that: (i) there are two or more competing $L$-dependent behaviors in $\mathbb{P}_\ell(L)$ for $\ell \geq 1$, and (ii) for sufficiently large $L$, $\mathbb{P}_\ell(L)$ would become a decreasing function of $L$.

### 5.1 Universal scaling form

We shall show that the leading $L$-dependent behavior of $\mathbb{P}_\ell(L)$ for any fixed $\ell \geq 1$ is described asymptotically by a universal scaling function that includes a logarithmic factor. First recall the definition of $\mathcal{W}_k(L)$ as the generating function of $\mathbb{P}_\ell(L)$, and its asymptotic $L$-dependent scaling form:

$$\mathcal{W}_k(L) = \sum_{\ell \geq 0} k^\ell \, \mathbb{P}_\ell(L), \tag{8}$$

$$\mathcal{W}_k(L) \simeq a_k \, L^{-X(k)}, \tag{9}$$

where both the NP exponent $X(k)$ and the non-universal constant $a_k$ are smooth functions of $k$ for $k > -1$. Let us also recall the scaling of $\mathbb{P}_0$ as obtained by setting $k = 0$ in Eqs. (8) and (9). Only the leading term, with $\ell = 0$, survives on the right-hand side (r.h.s.) of Eq. (8), and one has $\mathbb{P}_0 = \mathcal{W}_0 \sim a_0 L^{-1/4}$.

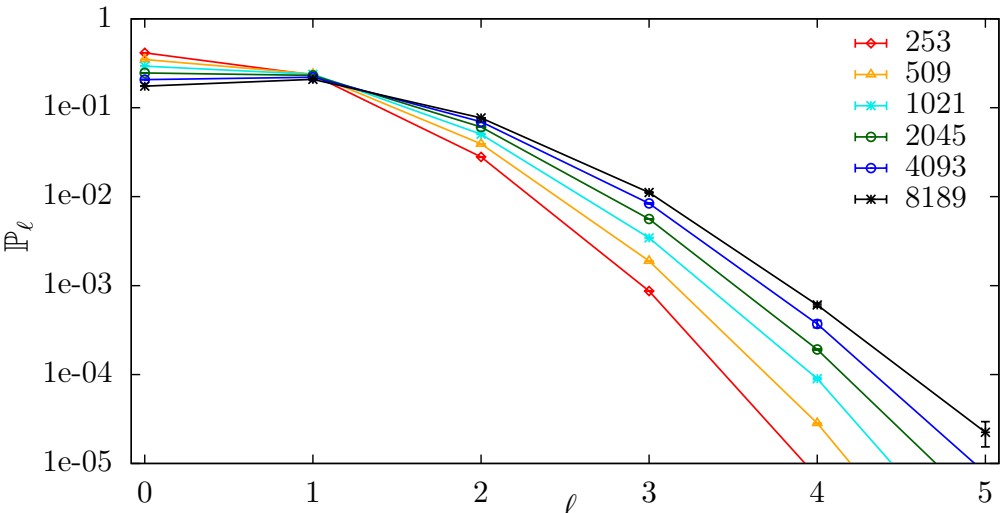

Figure 13: Probability distribution $\mathbb{P}_\ell$ versus the nested-path number $\ell$ for a series of sizes $L$ for STr at criticality.

Let us now derive the scaling of $\mathbb{P}_1$ by calculating the partial derivative of $\mathcal{W}_k$, with respective to $k$, and then setting $k = 0$. From Eqs. (8) and (9), we have

$$\frac{\partial \mathcal{W}_k}{\partial k} = \sum_{\ell \geq 1} \ell k^{\ell-1} \mathbb{P}_\ell \, , \tag{10}$$

$$\frac{\partial \mathcal{W}_k}{\partial k} \simeq (-X_k' \ln L) a_k L^{-X_k} + a_k' L^{-X_k}$$

$$= (-X_k' \ln L) \mathcal{W}_k [1 + \mathcal{O}(1/\ln L)] \, , \tag{11}$$

where the derivative of $L^{-X_k}$, with respect to $k$, gives a multiplicative logarithmic factor $\ln L$ and a universal amplitude $-X_k'$. The second term in the first line of Eq. (11) acts as a logarithmic subleading correction.

Similarly, by setting $k = 0$, only the term with $\ell = 1$, which is $\mathbb{P}_1(L)$, survives on the r.h.s. of Eq. (10), giving $\mathcal{W}_0' = \mathbb{P}_1$. From Eq. (11), we have

$$\mathbb{P}_1(L) \simeq a_0 L^{-1/4} (\Lambda \ln L) [1 + \mathcal{O}(1/\ln L)] \, , \tag{12}$$

where $\mathbb{P}_0 \simeq a_0 L^{-1/4}$ is used and the universal constant $\Lambda = -X_0' = 1/\sqrt{3}\pi$ can be calculated from Eq. (3).

The asymptotic scaling of $\mathbb{P}_\ell$ for $\ell > 1$ can be derived in an analogous way by taking the $\ell$-th derivative of $\mathcal{W}_k$ and setting $k = 0$. From Eqs. (8) and (9), we have

$$\frac{\partial^\ell \mathcal{W}_k}{\partial k^\ell} = \sum_{\ell' \geq \ell} \frac{(\ell')!}{(\ell' - \ell)!} k^{\ell'-\ell} \mathbb{P}_{\ell'} \tag{13}$$

$$\frac{\partial^\ell \mathcal{W}_k}{\partial k^\ell} \simeq (-X_k' \ln L)^\ell \mathcal{W}_k \left( 1 + \sum_{\ell'=1}^\ell \frac{b_{\ell'}}{(\ln L)^{\ell'}} \right) \, , \tag{14}$$

where a series of logarithmic subleading corrections arise. Setting $k = 0$ and combining these two equations give

$$\mathbb{P}_\ell \simeq a_0 L^{-1/4} \left[ \frac{1}{\ell!} (\Lambda \ln L)^\ell \right] \left( 1 + \sum_{\ell'=1}^\ell \frac{b_{\ell'}}{(\ln L)^{\ell'}} \right) \, . \tag{15}$$

Notice that, as $\ell$ increases, the scaling behaviors of $\mathbb{P}_\ell$ would involve a longer series of logarithmic corrections, which vanish extremely slowly.

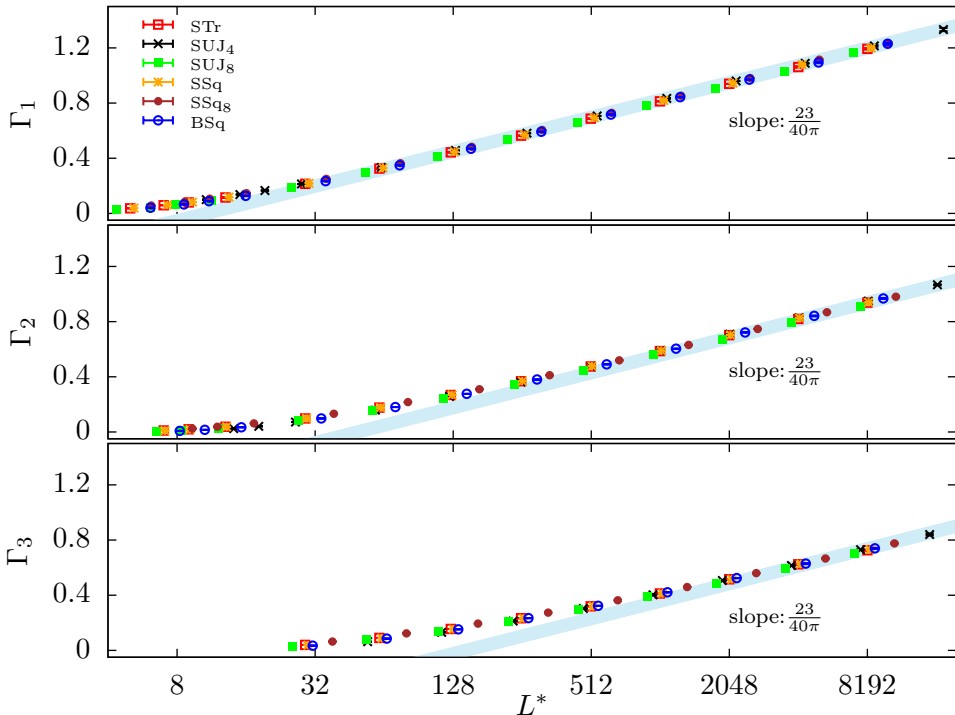

Figure 14: Semi-log plot of the ratios $\Gamma_1$, $\Gamma_2$ and $\Gamma_3$ versus re-scaled size $L^* = aL$, where $a$ is a model-dependent constant ($a = 1$ for STr). The asymptotic slope agrees well with the theoretical value $\Lambda = 1/\sqrt{3}\pi \approx 0.183\,776$.

| | $L_m$ | $\chi^2$/DF | $\Lambda$ | $b_1$ | $c_1$ | $c_2$ |
|---|---|---|---|---|---|---|
| STr | 29 | 0.17 | 0.1840(2) | -0.465(1) | 0.483(9) | 0.9(1) |
| | 61 | 0.19 | 0.1838(5) | -0.464(3) | 0.47(3) | 1.2(9) |
| $SUJ_4$ | 29 | 0.32 | 0.1845(4) | -0.319(2) | 0.24(2) | 0.9(2) |
| | 61 | 0.29 | 0.1840(7) | -0.316(5) | 0.21(5) | 2(1) |
| $SUJ_8$ | 29 | 1.11 | 0.1840(6) | -0.497(4) | 0.56(3) | 0.8(3) |
| | 61 | 1.15 | 0.184(1) | -0.494(9) | 0.52(9) | 2(2) |
| SSq | 29 | 0.33 | 0.1844(3) | -0.463(2) | 0.48(1) | 0.9(2) |
| | 61 | 0.41 | 0.1845(8) | -0.463(6) | 0.49(6) | 1(2) |
| $SSq_8$ | 29 | 0.91 | 0.1840(6) | -0.423(4) | 0.45(2) | 0.6(3) |
| | 61 | 1.10 | 0.184(1) | -0.425(9) | 0.5(1) | 0(2) |
| BSq | 29 | 0.22 | 0.1840(3) | -0.435(2) | 0.38(1) | 1.1(1) |
| | 61 | 0.24 | 0.1842(6) | -0.437(4) | 0.40(5) | 1(1) |

Table 4: Fitting results of $\Gamma_1$ by Eq. (16). For all the six percolation systems, the estimates for $\Lambda$ agree well with the predicted value $1/\sqrt{3}\pi \approx 0.183\,776$. The amplitude $b_1$ of the logarithmic correction is also well determined.

## 5.2 Numerical verification

In order to numerically verify the asymptotic universal scaling form (15), we consider the ratio $\Gamma_\ell \equiv (\ell!\mathbb{P}_\ell/\mathbb{P}_0)^{1/\ell}$ which, for any $\ell \geq 1$, should diverge as $\Gamma_\ell(L) \simeq \Lambda \ln L$ for $L \to \infty$. As mentioned above, finite-size corrections will become more severe as $\ell$ increases, which would obscure the numerical observation of the asymptotic logarithmic behavior, and therefore we do not consider $\Gamma_\ell$ with $\ell \geq 4$. The $\Gamma_\ell$ data with $\ell = 1, 2, 3$ are shown in Fig. 14, where the logarithmic divergence $\ln L$ and the universal amplitude $\Lambda = 1/\sqrt{3}\pi \approx 0.184$ are illustrated, and, indeed, the corrections for small $L$ are more pronounced for higher $\ell$.

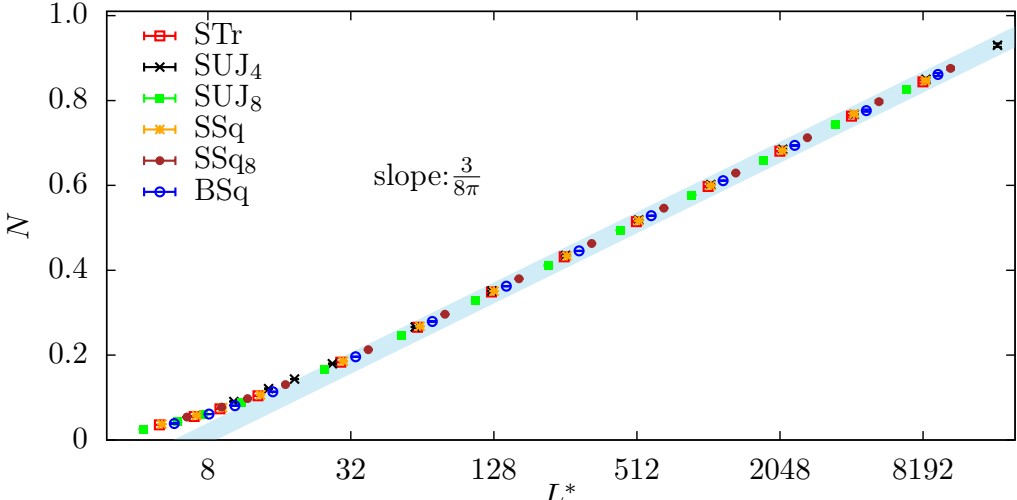

Figure 15: Semi-log plot of conditional nested-path number $N$ versus re-scaled system size $L^* = aL$, where $a$ is a model-dependent constant ($a = 1$ is for STr). The asymptotic slope agrees well with the theoretical value $\kappa = 3/8\pi \approx 0.119\,366$.

|       | $L_m$ | $\chi^2/\text{DF}$ | $\kappa$ | $b_1$ | $c_1$ | $c_2$ |
|-------|-------|--------------------|----------|-------|-------|-------|
| STr   | 29    | 0.55               | 0.1197(3)| -0.232(2)| 0.071(1) | 1.2(1) |
|       | 61    | 0.40               | 0.1194(4)| -0.230(3)| 0.04(3)  | 1.9(9) |
| SUJ$_4$ | 29  | 0.35               | 0.1197(2)| -0.145(2)| 0.05(1)  | 0.6(1) |
|       | 61    | 0.22               | 0.1193(4)| -0.142(3)| 0.02(3)  | 1.4(8) |
| SUJ$_8$ | 29  | 1.04               | 0.1198(3)| -0.254(2)| 0.09(1)  | 1.3(2) |
|       | 61    | 0.18               | 0.1193(3)| -0.250(2)| 0.04(2)  | 2.6(6) |
| SSq   | 29    | 0.53               | 0.1198(3)| -0.231(2)| 0.08(1)  | 1.1(1) |
|       | 61    | 0.44               | 0.1195(5)| -0.229(3)| 0.05(4)  | 1.8(9) |
| SSq$_8$ | 29  | 0.31               | 0.1196(2)| -0.200(1)| 0.050(9) | 1.0(1) |
|       | 61    | 0.38               | 0.1196(4)| -0.199(3)| 0.05(3)  | 1.1(9) |
| BSq   | 29    | 0.16               | 0.1196(2)| -0.218(1)| 0.068(7) | 0.80(8)|
|       | 61    | 0.14               | 0.1194(3)| -0.217(2)| 0.05(2)  | 1.2(5) |

Table 5: Fits of the conditional NP number $N$ by Eq. (17). The estimate of $\kappa$ is well consistent with the theoretical value $\kappa = 3/8\pi \approx 0.119\,366$.

579     We then fit the $\Gamma_\ell$ data to

$$\Gamma_\ell(L) = \ln L \left( \Lambda + \frac{b_1}{\ln L} + \frac{c_1}{L} + \frac{c_2}{L^2} \right), \tag{16}$$

580   which includes only the leading logarithmic correction term for simplicity, but includes con-
581   ventional correction terms, $1/L$ and $1/L^2$. The results for $\Gamma_1$ are given in Table 4. For all the
582   six percolation systems, the estimates of $\Lambda$ are in good agreement with the theoretical value
583   $\Lambda = 1/\sqrt{3}\pi$.

584     The correction term $b_1/\ln L$ is also well determined in Table 4, where the amplitude $b_1$ is
585   negative for all the systems and has similar magnitude. As seen from the first line in Eq. (11),
586   this logarithmic correction comes from the sub-leading term $a_k' L^{-X_k}$ at $k = 0$. Since $\Lambda = -X_0'$
587   and $a_0$ are both positive, the sign of $b_1$ must stem from the sign of $a_0'$. In other words, the
588   fitting results in Table 4 suggest that, near $k = 0$, the amplitude $a_k$ in the scaling $\mathcal{W}_k \simeq a_k L^{-X_k}$
589   is a decreasing function of $k$. Actually, for the whole range $k > -1$, $a_k$ is a monotonically de-
590   creasing function of $k$, as shown in Tables 9, 10 and 11 in the appendix where $a_k$ corresponds
591   to parameter $c_0$.

### 5.3 Discussion of logarithmic subleading corrections

In probability theory, if the probability distribution of some random variable $\{\mathcal{Y}\}$ is known, it is usually straightforward to derive the behavior of quantities that are defined in terms of $\{\mathcal{Y}\}$. However, for the current case for nested paths, this procedure does not work since the scaling of the probability distribution $\mathbb{P}_\ell$ is itself obtained from the scaling of the correlator $\mathcal{W}_k(L)$ near $k = 0$. As a consequence, the $L$-dependent scaling behavior of $\mathcal{W}_k(L)$ for $k \neq 0$ cannot be calculated from the asymptotic leading behavior of $\mathbb{P}_\ell(L)$ in Eq. (15). Take the percolating probability as an example, which, by definition, is $\langle \mathcal{R} \rangle = \mathcal{W}_1 \equiv \sum_{\ell \geq 0} \mathbb{P}_\ell(L)$. Notice that, for any $\ell \geq 1$, $\mathbb{P}_\ell$ involves the summation of $\ell + 1$ terms as $\mathbb{P}_\ell \simeq L^{-1/4} \sum_{\ell' \geq 0}^{\ell} b_{\ell'} (\ln L)^{\ell'}$. It seems impossible to obtain the pure power-law scaling $\mathcal{W}_1 \simeq L^{-5/48}$ from Eq. (15), unless all the logarithmic corrections are taken into account in a smart way.

In other words, we appear to be in a situation of non-commuting limits. Indeed, in the correlator $\mathcal{W}_k$ with a given $L$, the contributions from all possible $\ell$ are summed up, whereas Eq. (15) describes, for a fixed and finite $\ell$, the $L$-dependent scaling of $\mathbb{P}_\ell(L)$.

### 5.4 Conditional nested-path number

We now consider $\mathcal{W}_k$ and its derivative at $k = 1$. First of all, setting $k = 1$ in Eqs. (8) and (9) gives $\langle \mathcal{R} \rangle = \mathcal{W}_1 \sim L^{-X_1}$ with $X_1 = 5/48$. Then, for the first derivative, setting $k = 1$ in Eq. (10) leads to $\sum_{\ell \geq 0} \ell \, \mathbb{P}_\ell \equiv \langle \mathcal{R} \cdot \ell \rangle$, where the percolating indicator $\mathcal{R}$ ensures no contribution from non-percolating configurations. Further, from Eq. (11), we obtain $\langle \mathcal{R} \cdot \ell \rangle \simeq (\kappa \ln L) \mathcal{W}_1 [1 + \mathcal{O}(1/\ln L)]$, with $\kappa = -X_1' = 3/8\pi$.

In Monte Carlo simulations, it is convenient to define and sample $N \equiv \langle \mathcal{R} \cdot \ell \rangle / \langle \mathcal{R} \rangle$. Physically, $N$ represents the number of independent nested paths averaged in the ensemble of percolating configurations, and we shall call $N$ the conditional NP number. From the discussion above, $N$ is known to diverge logarithmically as $N \simeq \kappa \ln L [1 + \mathcal{O}(1/\ln L)]$. The data for $N$ in the six percolation models are shown in Fig. 15. Since the logarithmic scaling behavior is manifest, we fit the data to the form

$$N = \ln L \left( \kappa + \frac{b_1}{\ln L} + \frac{c_1}{L} + \frac{c_2}{L^2} \right), \tag{17}$$

and the results are given in Table 5. The estimate of $\kappa$ agrees well with the theoretical value $3/8\pi \approx 0.119\,366$.

The scaling behavior of the conditional NP number implies that, given any critical percolation cluster with gyration radius $r$, the mean number of nested paths diverges logarithmically as $\kappa \ln r$.

Similar calculations, involving higher-order derivatives of $\mathcal{W}_k$ at $k = 1$, imply that the number $N$ of nested paths is asymptotically normal, with average $\kappa \ln L$ (as stated above), and variance $\kappa' \ln L$, where $\kappa' = 3(\pi - 1)/(8\pi^2)$.

The probability distribution of PNPs and NLs can be obtained readily from the probability distribution $\mathbb{P}_\ell$ for MNPs. First recall the identities between the one point functions (3, 4): $W_{\text{PNP}}(k) = W_{\text{MNP}}(2k)$ and $W_{\text{NL}}(k) = W_{\text{MNP}}(k + 1)$. Expressing the one-point functions in the corresponding probability distributions, immediately leads to the following equations for the probability distributions of nested paths and loops, the type being indicated with a superscript:

$$\mathbb{P}_\ell^{\text{PNP}} = 2^\ell \, \mathbb{P}_\ell^{\text{MNP}} \quad \text{and} \quad \mathbb{P}_\ell^{\text{NL}} = \sum_{\ell' \geq \ell} \binom{\ell'}{\ell} \mathbb{P}_{\ell'}^{\text{MNP}}.$$

## 6  Discussion

Following the initial work [34] we have further studied the nested-path (NP) operator for two-dimensional critical percolation. We have complemented the original monochromatic version with a polychromatic variety. And we have derived analytical formulae (3)–(4) for the corresponding power-law exponents $X_{\mathrm{MNP}}(k)$ and $X_{\mathrm{PNP}}(k)$. By simulating six different percolation models, we have provided explicit and strong evidence for the universality of the power-law scaling, with respect to the linear size $L$, of the NP correlation function $\mathcal{W}_k(L)$. The fitting results of exponent $X_{\mathrm{MNP}}(k)$ are in excellent agreement with the formula (3) for a broad range of $k$. For the marginal case $k = -1$ with $X_{\mathrm{MNP}}(-1) = 2/3$, we have conjectured that the power-law scaling is modified by a multiplicative logarithmic correction as $\mathcal{W}_{-1} \sim L^{-2/3}(\ln L)$, which is also supported by our high-precision data.

For the $k = 2$ case, the exact identity $\mathcal{W}_2(L) = 1$ for site percolation on self-matching planar triangulation lattices has been well demonstrated for triangular and Union-Jack lattices with different center locations and domain shapes. However, for bond percolation on the square lattice, the identity $\mathcal{W}_2(L) = 1$ fails, and the asymptotic value of $\mathcal{W}_2(L \to \infty)$ depends on the domain shape and on the location of the center. Similarly, for SSq and SSq$_8$, an asymptotically matching pair of site percolation, neither the $\mathcal{W}_2(L \to \infty)$ values nor their average is equal to 1.

For the probability distributions $\mathbb{P}_\ell(L)$, we have derived the asymptotic $L$-dependent scaling (15), for any fixed and finite $\ell$, with the universal constant $\Lambda = 1/\sqrt{3}\pi$. In addition, we have shown that the conditional NP number $N$ diverges logarithmically as $N \simeq \kappa \ln L$, with $\kappa = 3/8\pi$. Excellent agreement between the numerical and theoretical results was observed, both for the probability ratios $\Gamma_\ell$ and the conditional NP number $N$.

Future work will consider the nested-path and nested-loop operators for other statistical-mechanical models in two dimensions, particularly the $Q$-state Potts model in the Fortuin-Kasteleyn cluster representation that includes bond percolation as a special case for $Q \to 1$.

## Acknowledgements

We (JLJ, BN, AS) acknowledge hospitality and support of the Galileo Galilei Institute in Firenze during the program "Randomness, Integrability, and Universality", where part of the collaboration of this project took place, and essential progress was made.

**Funding information**   The work of J.-F. S. and Y. D. was supported by the National Natural Science Foundation of China (under Grant No. 12275263) and Natural Science Foundation of Fujian province of China (under Grant No. 2023J02032) The work of J. L. J. was supported by the French Agence Nationale de la Recherche (ANR) under grant ANR-21-CE40-0003 (project CONFICA) and the European Research Council (under the Advanced Grant NuQFT). The work of A. S. was supported by the French Agence Nationale de la Recherche (ANR) under grants ANR-18-CE40-0033 (project DIMERS) and ANR-19-CE48-0011 (project COMBINE).

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

## A  Data for nested-path probability distribution $\mathbb{P}_\ell(L)$

Tables 6, 7 and 8 give the Monte Carlo data for the probability distribution $\mathbb{P}_\ell$, for the event that the center cluster is percolating ($\mathcal{R} = 1$) and has $\ell$ independent closed nested paths (NPs) surrounding the center, for all the six critical percolation models discussed in the main text. Let us recall our abbreviations for their names: bond percolation on the square lattice (BSq), and site percolation on the triangular (STr), Union-Jack (SUJ$_4$ and SUJ$_8$) and square lattice without/with next-nearest-neighbouring interactions (SSq and SSq$_8$). Also included are the data for the probability that the center cluster is not percolating ($\mathcal{R} = 0$). As the system size $L$ increases, the probability for $\mathcal{R} = 0$ grows and saturates to 1 as $1 - \mathcal{R} = 1 - aL^{-5/48}$, with $a$ a non-universal constant.

For the $\ell = 0$ case, the probability $\mathbb{P}_0$ monotonically vanishes as $L^{-1/4}$. However, the probability $\mathbb{P}_0$ keeps growing until $L = 509$, and then starts to decrease. This is due to the competing terms $\ln L$ and $L^{-1/4}$ in the scaling $\mathbb{P}_1(L) \propto (\ln L)L^{-1/4}$. For higher $\ell > 1$, since $\mathbb{P}_\ell(L) \propto (\ln L)^\ell L^{-1/4}$, the probability $\mathbb{P}_\ell(L)$ would keep increasing till even larger system size before it starts to drop.

## B  Fitting results of nested-path correlation function $\mathcal{W}_k(L)$

The results of fitting the nested-path (NP) correlation function $\mathcal{W}_k(L)$ by Eq. (7) are given in Tables 9, 10 and 11, where $L_m$ represents the cut-off linear size such that only the data for $L \geq L_m$ are admitted in the fits. As $k$ becomes negative and approaches $-1$, finite-size corrections become more and more severe, and the reliability of the fitting results decreases. Actually, for $k = -1$, we conjecture that Eq. (7) is modified by a multiplicative logarithmic correction. The estimated values of $X_{\mathrm{NP}}$ are consistent with each other for the six percolation models, and this strongly supports the universality of the nested-path operator.

| | $L$ | $\mathcal{R}=0$ | $\ell=0$ | 1 | 2 | 3 | 4 | 5 |
|---|---|---|---|---|---|---|---|---|
| | 3 | 0.015625(2) | 0.968748(2) | 0.015627(2) | | | | |
| | 5 | 0.034370(2) | 0.931273(4) | 0.034353(3) | 3.79(2)E-6 | | | |
| | 7 | 0.052561(2) | 0.894969(3) | 0.052425(3) | 4.54(1)E-5 | | | |
| | 9 | 0.068618(3) | 0.863076(6) | 0.068153(4) | 1.527(2)E-4 | 1.6(4)E-9 | | |
| | 13 | 0.094857(3) | 0.811400(4) | 0.093187(4) | 5.556(4)E-4 | 8.1(4)E-8 | | |
| | 29 | 0.157898(4) | 0.690883(7) | 0.147895(5) | 0.0033168(9) | 7.10(4)E-6 | 1.6(4)E-9 | |
| | 61 | 0.217367(5) | 0.583879(6) | 0.189584(4) | 0.009099(1) | 7.060(9)E-5 | 8.4(3)E-8 | |
| STr | 125 | 0.27252(2) | 0.49205(2) | 0.21752(2) | 0.01760(1) | 3.05(1)E-4 | 1.14(7)E-6 | 4(4)E-9 |
| | 253 | 0.32354(3) | 0.41418(3) | 0.23340(2) | 0.02801(1) | 8.67(2)E-4 | 7.1(2)E-6 | 8(5)E-9 |
| | 509 | 0.37089(3) | 0.34844(3) | 0.23953(2) | 0.039222(9) | 0.001889(2) | 2.83(4)E-5 | 1.5(2)E-7 |
| | 1021 | 0.4147(1) | 0.2932(1) | 0.2382(1) | 0.05032(5) | 0.00344(1) | 9.0(2)E-5 | 6(2)E-7 |
| | 2045 | 0.4557(1) | 0.2464(1) | 0.23142(8) | 0.06071(5) | 0.00561(2) | 1.91(3)E-4 | 2.9(4)E-6 |
| | 4093 | 0.4943(7) | 0.2073(7) | 0.2201(6) | 0.0695(4) | 0.0084(1) | 3.7(3)E-4 | 8(4)E-6 |
| | 8189 | 0.5278(7) | 0.1749(4) | 0.2086(5) | 0.0769(3) | 0.0111(2) | 6.1(3)E-4 | 2.2(7)E-5 |
| | | | | | | | | |
| | 3 | 0.015624(2) | 0.968750(3) | 0.015626(2) | | | | |
| | 5 | 0.037387(2) | 0.925263(3) | 0.037349(2) | 9.3(1)E-7 | | | |
| | 7 | 0.057693(3) | 0.884854(4) | 0.057429(4) | 2.412(6)E-5 | | | |
| | 9 | 0.075222(4) | 0.850168(6) | 0.074508(5) | 1.027(1)E-4 | | | |
| | 13 | 0.103291(3) | 0.795233(5) | 0.101025(4) | 4.514(3)E-4 | 2.1(2)E-8 | | |
| | 29 | 0.168562(8) | 0.671663(8) | 0.156571(5) | 0.0032001(8) | 4.37(4)E-6 | | |
| BSq | 61 | 0.228468(6) | 0.565362(7) | 0.196889(5) | 0.009223(1) | 5.757(9)E-5 | 3.8(3)E-8 | |
| | 125 | 0.28335(3) | 0.47551(3) | 0.22277(3) | 0.018093(8) | 2.80(1)E-4 | 7.7(6)E-7 | |
| | 253 | 0.33376(3) | 0.39994(4) | 0.23660(3) | 0.02885(1) | 8.45(2)E-4 | 6.0(2)E-6 | 4(4)E-9 |
| | 509 | 0.38050(4) | 0.33631(4) | 0.24095(3) | 0.040331(9) | 0.001887(3) | 2.54(3)E-5 | 1.0(2)E-7 |
| | 1021 | 0.4237(1) | 0.28289(8) | 0.2383(1) | 0.05150(5) | 0.00352(1) | 7.7(2)E-5 | 5(1)E-7 |
| | 2045 | 0.4641(1) | 0.23773(9) | 0.23045(9) | 0.06187(5) | 0.00568(2) | 1.94(4)E-4 | 2.2(4)E-6 |
| | 4093 | 0.5008(7) | 0.2004(6) | 0.2192(5) | 0.0709(4) | 0.0083(1) | 3.9(3)E-4 | 2(2)E-6 |
| | 8189 | 0.5365(8) | 0.1673(5) | 0.2059(7) | 0.0783(4) | 0.0113(2) | 7.0(3)E-4 | 2.0(5)E-5 |

Table 6: Monte Carlo data for nested-path probability distribution $\mathbb{P}_\ell(L)$ for STr and BSq. The column with $\mathcal{R}=0$ represents the probability that the center cluster is not percolating. The maximum number of nested paths is $\ell_{\max}=(L-1)/2$ ($L$ is odd), which is $1,2,3$ respectively for $L=3,5,7$. However, due to the super-exponentially fast decaying of $\mathbb{P}_\ell(L)$ as $\ell$ increases, the probability $\mathbb{P}_2(L=5)$ is already very small, which is O($10^{-6}$) for BSq and similar for the others.

| | $L$ | $\mathcal{R}=0$ | $\ell=0$ | 1 | 2 | 3 | 4 | 5 |
|---|---|---|---|---|---|---|---|---|
| SSq | 3 | 0.027508(2) | 0.957254(2) | 0.015239(2) | | | | |
| | 5 | 0.052157(3) | 0.912493(4) | 0.035346(2) | 3.57(3)E-6 | | | |
| | 7 | 0.073218(4) | 0.873314(5) | 0.053427(3) | 4.07(1)E-5 | | | |
| | 9 | 0.090746(4) | 0.840281(5) | 0.068834(4) | 1.396(2)E-4 | 1.0(4)E-9 | | |
| | 13 | 0.118292(4) | 0.788115(6) | 0.093074(4) | 5.188(3)E-4 | 5.4(3)E-8 | | |
| | 29 | 0.181852(5) | 0.669222(6) | 0.145745(4) | 0.0031753(9) | 6.04(3)E-6 | 2(2)E-10 | |
| | 61 | 0.240394(6) | 0.564957(8) | 0.185786(5) | 0.008799(1) | 6.383(9)E-5 | 6.5(3)E-8 | |
| | 125 | 0.29424(2) | 0.47585(3) | 0.21251(2) | 0.017111(9) | 2.88(1)E-4 | 8.8(5)E-7 | |
| | 253 | 0.34382(2) | 0.40048(3) | 0.22761(3) | 0.02725(1) | 8.27(2)E-4 | 6.4(1)E-6 | 1.6(7)E-8 |
| | 509 | 0.38978(3) | 0.33685(3) | 0.23328(3) | 0.03825(1) | 0.001816(2) | 2.61(3)E-5 | 1.1(2)E-7 |
| | 1021 | 0.43235(9) | 0.28328(8) | 0.23187(7) | 0.04910(5) | 0.00331(1) | 8.0(2)E-5 | 3(1)E-7 |
| | 2045 | 0.4720(1) | 0.23826(8) | 0.2251(1) | 0.05902(5) | 0.00543(1) | 1.86(4)E-4 | 2.0(3)E-6 |
| | 4093 | 0.5075(6) | 0.2000(4) | 0.2156(6) | 0.0683(3) | 0.0082(1) | 3.5(3)E-4 | 1.8(5)E-5 |
| | 8189 | 0.5431(8) | 0.1685(5) | 0.2022(6) | 0.0745(4) | 0.0109(1) | 7.0(5)E-4 | 2.0(4)E-5 |
| SSq$_8$ | 3 | 0.015240(2) | 0.957247(2) | 0.027513(2) | | | | |
| | 5 | 0.035373(3) | 0.912497(4) | 0.052081(3) | 4.94(1)E-5 | | | |
| | 7 | 0.053661(3) | 0.873305(4) | 0.072780(3) | 2.541(3)E-4 | 6(1)E-9 | | |
| | 9 | 0.069497(4) | 0.840293(5) | 0.089621(3) | 5.890(3)E-4 | 1.00(5)E-7 | | |
| | 13 | 0.095236(3) | 0.788110(5) | 0.115157(4) | 0.0014962(6) | 1.19(2)E-6 | | |
| | 29 | 0.157254(4) | 0.669231(5) | 0.167651(6) | 0.0058350(8) | 2.926(7)E-5 | 1.6(1)E-8 | |
| | 61 | 0.216242(8) | 0.564967(7) | 0.205464(6) | 0.013154(2) | 1.729(2)E-4 | 4.40(9)E-7 | 6(3)E-10 |
| | 125 | 0.27126(3) | 0.47585(3) | 0.22949(2) | 0.022821(9) | 5.74(2)E-4 | 3.52(9)E-6 | 4(4)E-9 |
| | 253 | 0.32222(3) | 0.40050(3) | 0.24204(3) | 0.03386(1) | 0.001370(3) | 1.70(3)E-5 | 5(1)E-8 |
| | 509 | 0.36961(3) | 0.33690(3) | 0.24543(3) | 0.04533(2) | 0.002685(3) | 5.35(4)E-5 | 3.8(4)E-7 |
| | 1021 | 0.4135(1) | 0.2835(1) | 0.2419(1) | 0.05635(3) | 0.00457(2) | 1.40(2)E-4 | 1.6(3)E-6 |
| | 2045 | 0.45438(9) | 0.23822(9) | 0.2338(1) | 0.06637(6) | 0.00695(3) | 2.94(4)E-4 | 4.8(5)E-6 |
| | 4093 | 0.4916(8) | 0.1999(6) | 0.2228(6) | 0.0753(4) | 0.0098(1) | 5.8(3)E-4 | 1.4(6)E-5 |
| | 8189 | 0.5285(9) | 0.1686(6) | 0.2077(6) | 0.0811(4) | 0.0131(2) | 8.9(5)E-4 | 2.7(8)E-5 |

Table 7: Monte Carlo data for nested-path probability distribution $\mathbb{P}_\ell(L)$ for SSq and SSq$_8$.

| | $L$ | $\mathcal{R}=0$ | $\ell=0$ | 1 | 2 | 3 | 4 | 5 |
|---|---|---|---|---|---|---|---|---|
| | 3 | 0.062497(3) | 0.875000(5) | 0.062503(3) | | | | |
| | 5 | 0.083185(4) | 0.833667(6) | 0.083133(4) | 1.520(4)E-5 | | | |
| | 7 | 0.107972(3) | 0.784486(5) | 0.107327(4) | 2.157(1)E-4 | | | |
| | 9 | 0.126083(6) | 0.749038(8) | 0.124273(4) | 6.061(4)E-4 | 8(1)E-9 | | |
| | 13 | 0.153895(7) | 0.695711(9) | 0.148642(6) | 0.0017513(6) | 3.64(9)E-7 | | |
| | 29 | 0.215997(6) | 0.582639(7) | 0.194085(5) | 0.007255(1) | 2.375(6)E-5 | 3.1(7)E-9 | |
| $\mathrm{SUJ}_4$ | 61 | 0.272319(6) | 0.488592(8) | 0.222841(3) | 0.016065(2) | 1.821(1)E-4 | 2.85(7)E-7 | |
| | 125 | 0.32387(2) | 0.41022(3) | 0.23827(2) | 0.02699(1) | 6.54(2)E-4 | 3.2(1)E-6 | 4(4)E-9 |
| | 253 | 0.37145(3) | 0.34460(3) | 0.24355(2) | 0.03878(1) | 0.001604(3) | 1.70(3)E-5 | 4(1)E-8 |
| | 509 | 0.41543(3) | 0.28961(3) | 0.24136(2) | 0.05039(1) | 0.003150(2) | 5.99(4)E-5 | 3.4(5)E-7 |
| | 1021 | 0.4562(1) | 0.24350(9) | 0.2337(1) | 0.06109(5) | 0.00527(2) | 1.58(4)E-4 | 1.3(2)E-6 |
| | 2045 | 0.4943(1) | 0.20465(7) | 0.22264(9) | 0.07015(5) | 0.00794(2) | 3.40(4)E-4 | 6.5(6)E-6 |
| | 4093 | 0.5298(7) | 0.1721(4) | 0.2089(5) | 0.0774(3) | 0.0111(1) | 6.2(4)E-4 | 1.6(7)E-5 |
| | 8189 | 0.5631(8) | 0.1452(6) | 0.1936(9) | 0.0826(3) | 0.0143(3) | 0.00109(5) | 4(1)E-5 |
| | 3 | 0.0039074(7) | 0.992185(1) | 0.0039076(8) | | | | |
| | 5 | 0.024692(2) | 0.950617(2) | 0.024691(2) | 5.9(2)E-8 | | | |
| | 7 | 0.041136(3) | 0.917751(4) | 0.041104(2) | 8.98(4)E-6 | | | |
| | 9 | 0.056485(4) | 0.887134(6) | 0.056327(3) | 5.39(1)E-5 | | | |
| | 13 | 0.081778(4) | 0.837026(4) | 0.080906(3) | 2.906(3)E-4 | 1.1(2)E-8 | | |
| | 29 | 0.143930(5) | 0.716968(8) | 0.136699(6) | 0.0024002(7) | 3.11(2)E-6 | 2(2)E-10 | |
| $\mathrm{SUJ}_8$ | 61 | 0.203653(6) | 0.607812(9) | 0.181063(6) | 0.007429(1) | 4.358(9)E-5 | 2.8(3)E-8 | |
| | 125 | 0.25950(3) | 0.51304(4) | 0.21193(3) | 0.015308(4) | 2.221(8)E-4 | 5.2(4)E-7 | |
| | 253 | 0.31139(3) | 0.43210(2) | 0.23052(3) | 0.025306(9) | 6.85(1)E-4 | 4.6(1)E-6 | 2(1)E-8 |
| | 509 | 0.35950(3) | 0.36370(3) | 0.23876(3) | 0.03644(1) | 0.001579(2) | 2.03(2)E-5 | 6(2)E-8 |
| | 1021 | 0.4040(1) | 0.3059(1) | 0.23928(8) | 0.04769(3) | 0.00300(1) | 6.3(2)E-5 | 4(1)E-7 |
| | 2045 | 0.4457(1) | 0.25741(9) | 0.23356(9) | 0.05819(5) | 0.00499(1) | 1.62(3)E-4 | 2.1(4)E-6 |
| | 4093 | 0.4845(8) | 0.2166(6) | 0.2234(4) | 0.0676(3) | 0.0076(1) | 3.4(3)E-4 | 6(3)E-6 |
| | 8189 | 0.5190(8) | 0.1821(7) | 0.2128(4) | 0.0750(4) | 0.0105(1) | 6.3(4)E-4 | 2.9(8)E-5 |

Table 8: Monte Carlo data for nested-path probability distribution $\mathbb{P}_\ell(L)$ for $\mathrm{SUJ}_4$ and $\mathrm{SUJ}_8$.

| | $k$ | $L_m$ | $-X_{NP}$ | $c_0$ | $c_1$ | | $k$ | $L_m$ | $-X_{NP}$ | $c_0$ | $c_1$ |
|---|---|---|---|---|---|---|---|---|---|---|---|
| | 57.75 | 29 | 1.31(1) | 0.25(2) | 0.3(1) | | 5.02 | 29 | 0.2163(6) | 0.726(3) | 0.18(8) |
| | 52.12 | 29 | 1.25(1) | 0.26(2) | 0.3(1) | | 3.88 | 29 | 0.1461(4) | 0.799(2) | 0.13(6) |
| | 46.90 | 29 | 1.19(1) | 0.27(1) | 0.3(1) | | 2.88 | 29 | 0.0742(3) | 0.889(2) | 0.07(5) |
| | 42.09 | 29 | 1.130(9) | 0.28(1) | 0.4(1) | | 2.00 | 29 | 0.0000(2) | 1.000(1) | 0.00(4) |
| | 37.65 | 29 | 1.068(7) | 0.30(1) | 0.36(8) | | 1.60 | 29 | -0.0383(2) | 1.068(1) | -0.05(3) |
| | 33.56 | 29 | 1.005(6) | 0.315(9) | 0.37(7) | | 1.23 | 29 | -0.0775(1) | 1.146(1) | -0.14(3) |
| | 29.80 | 29 | 0.941(5) | 0.333(8) | 0.38(6) | | 1.00 | 29 | -0.1043(1) | 1.206(1) | -0.22(3) |
| | 26.35 | 29 | 0.877(4) | 0.353(6) | 0.38(5) | | 0.89 | 29 | -0.1179(1) | 1.238(1) | -0.27(3) |
| STr | 23.18 | 29 | 0.813(3) | 0.375(5) | 0.38(4) | | 0.57 | 29 | -0.1599(2) | 1.349(2) | -0.50(4) |
| | 20.29 | 29 | 0.749(2) | 0.398(4) | 0.38(4) | | 0.27 | 29 | -0.2037(2) | 1.485(2) | -0.91(6) |
| | 17.66 | 29 | 0.684(2) | 0.425(3) | 0.37(3) | | -0.00 | 29 | -0.2500(3) | 1.659(3) | -1.64(8) |
| | 15.26 | 29 | 0.619(1) | 0.454(3) | 0.36(2) | | -0.25 | 29 | -0.2994(4) | 1.885(4) | -3.0(1) |
| | 13.08 | 29 | 0.5532(9) | 0.486(2) | 0.35(2) | | -0.48 | 61 | -0.354(1) | 2.21(2) | -6.5(9) |
| | 11.11 | 29 | 0.4872(7) | 0.522(2) | 0.33(2) | | -0.69 | 61 | -0.414(2) | 2.68(3) | -13(1) |
| | 9.33 | 29 | 0.4206(5) | 0.563(1) | 0.30(1) | | -0.88 | 61 | -0.483(2) | 3.42(5) | -28(3) |
| | 7.73 | 29 | 0.353(1) | 0.610(5) | 0.3(1) | | -1.00 | 125 | -0.544(6) | 4.5(2) | -78(17) |
| | 6.30 | 29 | 0.2853(9) | 0.663(4) | 0.2(1) | | | | | | |
| | | | | | | | | | | | |
| | 57.75 | 29 | 1.31(1) | 0.24(1) | 0.5(1) | | 5.02 | 61 | 0.215(2) | 0.738(9) | 0.0(4) |
| | 52.12 | 29 | 1.25(1) | 0.25(1) | 0.5(1) | | 3.88 | 61 | 0.145(1) | 0.808(7) | 0.0(3) |
| | 46.90 | 29 | 1.188(9) | 0.27(1) | 0.54(9) | | 2.88 | 61 | 0.0736(8) | 0.893(5) | 0.0(3) |
| | 42.09 | 29 | 1.126(7) | 0.28(1) | 0.55(8) | | 2.00 | 61 | -0.0004(5) | 0.997(4) | 0.0(2) |
| | 37.65 | 29 | 1.064(6) | 0.297(9) | 0.56(7) | | 1.60 | 61 | -0.0385(4) | 1.061(3) | -0.1(2) |
| | 33.56 | 29 | 1.001(5) | 0.314(7) | 0.56(6) | | 1.23 | 61 | -0.0777(4) | 1.133(3) | -0.2(2) |
| | 29.80 | 29 | 0.938(4) | 0.334(6) | 0.56(5) | | 1.00 | 61 | -0.1044(4) | 1.188(3) | -0.2(2) |
| | 26.35 | 29 | 0.874(3) | 0.355(5) | 0.56(4) | | 0.89 | 61 | -0.1180(4) | 1.218(3) | -0.3(2) |
| BSq | 23.18 | 29 | 0.810(2) | 0.378(4) | 0.56(4) | | 0.57 | 61 | -0.1600(4) | 1.320(4) | -0.5(2) |
| | 20.29 | 29 | 0.746(2) | 0.403(4) | 0.55(3) | | 0.27 | 61 | -0.2038(5) | 1.445(5) | -0.8(3) |
| | 17.66 | 29 | 0.682(1) | 0.430(3) | 0.55(2) | | -0.00 | 61 | -0.2503(6) | 1.603(7) | -1.5(4) |
| | 15.26 | 29 | 0.617(1) | 0.461(3) | 0.53(2) | | -0.25 | 61 | -0.3002(8) | 1.81(1) | -2.8(6) |
| | 13.08 | 29 | 0.5514(9) | 0.494(2) | 0.51(2) | | -0.48 | 61 | -0.355(1) | 2.11(2) | -5.5(9) |
| | 11.11 | 29 | 0.4856(7) | 0.531(2) | 0.49(1) | | -0.69 | 61 | -0.416(2) | 2.54(3) | -11(2) |
| | 9.33 | 29 | 0.4192(5) | 0.573(1) | 0.45(1) | | -0.88 | 125 | -0.493(5) | 3.4(1) | -38(12) |
| | 7.73 | 29 | 0.3522(4) | 0.620(1) | 0.41(1) | | -1.00 | 125 | -0.551(6) | 4.3(2) | -73(20) |
| | 6.30 | 61 | 0.283(2) | 0.68(1) | 0.0(5) | | | | | | |

Table 9: Fitting results of the nested-path correlation function $\mathcal{W}_k(L)$ for STr and BSq by Eq. (7). When $k$ is large, the fit is already good without the correction term with $c_2$ being taken into account, and for simplicity the amplitude $c_2$ is not presented.

|      | $k$   | $L_\mathrm{m}$ | $-X_\mathrm{NP}$ | $c_0$ | $c_1$ | | $k$ | $L_\mathrm{m}$ | $-X_\mathrm{NP}$ | $c_0$ | $c_1$ |
|------|-------|------|---------|---------|---------|---|-------|------|-----------|----------|----------|
|      | 57.75 | 61 | 1.29(3) | 0.27(4) | 0.0(8) | | 5.02  | 61 | 0.215(1)   | 0.712(7) | 0.0(4) |
|      | 52.12 | 61 | 1.23(2) | 0.27(4) | 0.1(7) | | 3.88  | 61 | 0.145(1)   | 0.781(6) | 0.0(3) |
|      | 46.90 | 61 | 1.18(2) | 0.28(3) | 0.1(6) | | 2.88  | 61 | 0.0738(7)  | 0.866(4) | 0.0(2) |
|      | 42.09 | 61 | 1.12(2) | 0.29(3) | 0.2(5) | | 2.00  | 61 | -0.0002(5) | 0.972(3) | 0.0(2) |
|      | 37.65 | 61 | 1.06(1) | 0.30(2) | 0.2(5) | | 1.60  | 61 | -0.0383(4) | 1.037(3) | -0.1(2) |
|      | 33.56 | 61 | 1.00(1) | 0.32(2) | 0.3(4) | | 1.23  | 61 | -0.0775(3) | 1.112(3) | -0.1(1) |
|      | 29.80 | 61 | 0.934(9) | 0.33(2) | 0.3(4) | | 1.00  | 61 | -0.1043(3) | 1.169(3) | -0.2(1) |
|      | 26.35 | 61 | 0.871(8) | 0.35(2) | 0.3(3) | | 0.89  | 61 | -0.1179(3) | 1.201(3) | -0.3(1) |
| SSq  | 23.18 | 61 | 0.808(6) | 0.37(1) | 0.3(3) | | 0.57  | 61 | -0.1598(4) | 1.307(3) | -0.5(2) |
|      | 20.29 | 61 | 0.744(5) | 0.39(1) | 0.3(2) | | 0.27  | 61 | -0.2037(4) | 1.438(4) | -0.8(2) |
|      | 17.66 | 61 | 0.67(1) | 0.46(3) | -2(2) | | -0.00 | 61 | -0.2502(5) | 1.605(6) | -1.6(3) |
|      | 15.26 | 61 | 0.607(8) | 0.48(3) | -1(1) | | -0.25 | 61 | -0.3002(7) | 1.829(9) | -3.2(5) |
|      | 13.08 | 61 | 0.544(6) | 0.50(2) | -1(1) | | -0.48 | 61 | -0.355(1)  | 2.14(2)  | -6.6(8) |
|      | 11.11 | 61 | 0.480(5) | 0.53(2) | -0.6(9) | | -0.69 | 61 | -0.416(1) | 2.61(3)  | -14(1) |
|      | 9.33  | 61 | 0.416(3) | 0.56(1) | -0.4(7) | | -0.88 | 125 | -0.486(2) | 3.34(5)  | -29(3) |
|      | 7.73  | 61 | 0.350(3) | 0.61(1) | -0.3(6) | | -1.00 | 125 | -0.549(6) | 4.5(2)   | -85(19) |
|      | 6.30  | 61 | 0.283(2) | 0.654(9) | -0.1(5) | | | | | | |
|      |       |    |         |         |         | | | | | | |
|      | 57.75 | 29 | 1.30(2) | 0.45(4) | -0.2(4) | | 5.02  | 61 | 0.216(2)   | 0.802(9) | 0.0(4) |
|      | 52.12 | 29 | 1.25(1) | 0.45(3) | -0.1(3) | | 3.88  | 61 | 0.146(1)   | 0.862(6) | 0.0(3) |
|      | 46.90 | 29 | 1.19(1) | 0.45(3) | -0.1(3) | | 2.88  | 61 | 0.0742(7)  | 0.937(4) | -0.1(2) |
|      | 42.09 | 29 | 1.12(1) | 0.46(2) | 0.0(2) | | 2.00  | 61 | 0.0000(4)  | 1.032(3) | -0.1(2) |
|      | 37.65 | 29 | 1.063(8) | 0.47(2) | 0.0(2) | | 1.60  | 61 | -0.0382(3) | 1.089(3) | -0.1(1) |
|      | 33.56 | 29 | 1.001(7) | 0.48(2) | 0.1(1) | | 1.23  | 61 | -0.0774(3) | 1.156(2) | -0.2(1) |
|      | 29.80 | 29 | 0.938(5) | 0.49(1) | 0.1(1) | | 1.00  | 61 | -0.1042(3) | 1.207(2) | -0.3(1) |
|      | 26.35 | 29 | 0.875(4) | 0.50(1) | 0.10(9) | | 0.89  | 61 | -0.1178(3) | 1.235(3) | -0.3(1) |
| $\mathrm{SSq_8}$ | 23.18 | 29 | 0.812(3) | 0.520(8) | 0.11(7) | | 0.57 | 61 | -0.1598(3) | 1.332(3) | -0.5(2) |
|      | 20.29 | 29 | 0.748(2) | 0.538(6) | 0.12(6) | | 0.27  | 61 | -0.2038(4) | 1.452(4) | -0.9(2) |
|      | 17.66 | 61 | 0.68(2) | 0.58(8) | -1(4) | | -0.00 | 61 | -0.2503(6) | 1.606(6) | -1.7(3) |
|      | 15.26 | 61 | 0.61(1) | 0.60(6) | -1(3) | | -0.25 | 61 | -0.3001(8) | 1.81(1)  | -3.2(5) |
|      | 13.08 | 61 | 0.55(1) | 0.62(5) | 0(2) | | -0.48 | 61 | -0.354(1)  | 2.10(2)  | -6.2(9) |
|      | 11.11 | 61 | 0.485(7) | 0.64(3) | 0(2) | | -0.69 | 61 | -0.415(2) | 2.52(3)  | -12(2) |
|      | 9.33  | 61 | 0.419(5) | 0.68(2) | 0(1) | | -0.88 | 61 | -0.484(2) | 3.18(5)  | -25(3) |
|      | 7.73  | 61 | 0.352(3) | 0.71(2) | -0.1(8) | | -1.00 | 125 | -0.548(7) | 4.2(2)   | -79(22) |
|      | 6.30  | 61 | 0.285(2) | 0.75(1) | 0.0(6) | | | | | | |

Table 10: Fitting results of the nested-path correlation function $\mathcal{W}_k(L)$ for SSq and $\mathrm{SSq_8}$ by Eq. (7).

|        | $k$   | $L_m$ | $-X_{NP}$ | $c_0$ | $c_1$ | $k$   | $L_m$ | $-X_{NP}$ | $c_0$ | $c_1$ |
|--------|-------|-------|-----------|-------|-------|-------|-------|-----------|-------|-------|
|        | 57.75 | 29    | 1.31(2)   | 0.47(4)   | 0.6(3)   | 5.02  | 29    | 0.2167(2)  | 0.8304(8) | 0.278(8)  |
|        | 52.12 | 29    | 1.25(2)   | 0.48(4)   | 0.7(3)   | 3.88  | 29    | 0.1463(1)  | 0.8772(6) | 0.199(6)  |
|        | 46.90 | 29    | 1.19(1)   | 0.49(3)   | 0.7(2)   | 2.88  | 29    | 0.0743(1)  | 0.9325(4) | 0.108(5)  |
|        | 42.09 | 29    | 1.13(1)   | 0.50(3)   | 0.7(2)   | 2.00  | 29    | 0.0000(3)  | 1.000(2)  | 0.00(5)   |
|        | 37.65 | 29    | 1.067(9)  | 0.51(2)   | 0.7(2)   | 1.60  | 29    | -0.0383(2) | 1.040(2)  | -0.06(5)  |
|        | 33.56 | 29    | 1.004(7)  | 0.53(2)   | 0.7(1)   | 1.23  | 29    | -0.0775(2) | 1.086(1)  | -0.12(4)  |
|        | 29.80 | 29    | 0.940(6)  | 0.54(2)   | 0.7(1)   | 1.00  | 29    | -0.1043(2) | 1.120(1)  | -0.18(4)  |
|        | 26.35 | 29    | 0.877(5)  | 0.56(1)   | 0.7(1)   | 0.89  | 29    | -0.1179(2) | 1.139(1)  | -0.20(4)  |
| $SUJ_4$ | 23.18 | 29   | 0.813(4)  | 0.58(1)   | 0.66(8)  | 0.57  | 29    | -0.1599(2) | 1.203(2)  | -0.31(4)  |
|        | 20.29 | 29    | 0.748(3)  | 0.599(8)  | 0.63(7)  | 0.27  | 29    | -0.2038(2) | 1.281(2)  | -0.48(6)  |
|        | 17.66 | 29    | 0.684(2)  | 0.620(6)  | 0.60(5)  | -0.00 | 29    | -0.2503(3) | 1.380(3)  | -0.78(8)  |
|        | 15.26 | 29    | 0.619(2)  | 0.643(5)  | 0.57(4)  | -0.25 | 29    | -0.3004(4) | 1.512(4)  | -1.3(1)   |
|        | 13.08 | 29    | 0.553(1)  | 0.667(4)  | 0.54(3)  | -0.48 | 61    | -0.356(1)  | 1.70(2)   | -2.7(8)   |
|        | 11.11 | 29    | 0.4873(8) | 0.693(3)  | 0.50(2)  | -0.69 | 61    | -0.419(2)  | 1.99(3)   | -6(1)     |
|        | 9.33  | 29    | 0.4208(6) | 0.722(2)  | 0.46(2)  | -0.88 | 125   | -0.499(6)  | 2.6(1)    | -24(10)   |
|        | 7.73  | 29    | 0.3537(4) | 0.754(2)  | 0.41(1)  | -1.00 | 125   | -0.564(9)  | 3.3(2)    | -50(19)   |
|        | 6.30  | 29    | 0.2857(3) | 0.790(1)  | 0.35(1)  |       |       |            |           |           |
|        |       |       |           |           |          |       |       |            |           |           |
|        | 57.75 | 29    | 1.31(2)   | 0.20(1)   | 0.3(1)   | 5.02  | 29    | 0.2165(2)  | 0.6980(7) | 0.236(7)  |
|        | 52.12 | 29    | 1.25(1)   | 0.21(1)   | 0.3(1)   | 3.88  | 29    | 0.1463(1)  | 0.7782(5) | 0.189(5)  |
|        | 46.90 | 29    | 1.19(1)   | 0.22(1)   | 0.3(1)   | 2.88  | 29    | 0.0743(1)  | 0.8762(4) | 0.120(4)  |
|        | 42.09 | 29    | 1.127(9)  | 0.24(1)   | 0.32(8)  | 2.00  | 29    | 0.0000(1)  | 1.0002(3) | -0.002(3) |
|        | 37.65 | 29    | 1.064(8)  | 0.250(9)  | 0.33(7)  | 1.60  | 61    | -0.0381(4) | 1.073(3)  | 0.0(2)    |
|        | 33.56 | 29    | 1.002(6)  | 0.267(8)  | 0.34(6)  | 1.23  | 61    | -0.0772(3) | 1.159(3)  | -0.1(1)   |
|        | 29.80 | 29    | 0.939(5)  | 0.285(7)  | 0.35(5)  | 1.00  | 61    | -0.1040(3) | 1.225(3)  | -0.1(2)   |
|        | 26.35 | 29    | 0.876(4)  | 0.304(6)  | 0.36(5)  | 0.89  | 61    | -0.1176(3) | 1.261(3)  | -0.2(2)   |
| $SUJ_8$ | 23.18 | 29   | 0.812(3)  | 0.326(5)  | 0.37(4)  | 0.57  | 61    | -0.1595(4) | 1.383(3)  | -0.4(2)   |
|        | 20.29 | 29    | 0.748(2)  | 0.351(4)  | 0.37(3)  | 0.27  | 61    | -0.2032(4) | 1.533(4)  | -0.8(2)   |
|        | 17.66 | 29    | 0.683(2)  | 0.378(3)  | 0.37(3)  | -0.00 | 61    | -0.2495(5) | 1.725(6)  | -1.7(4)   |
|        | 15.26 | 29    | 0.618(1)  | 0.408(2)  | 0.37(2)  | -0.25 | 61    | -0.2990(7) | 1.98(1)   | -3.5(5)   |
|        | 13.08 | 29    | 0.5528(9) | 0.442(2)  | 0.36(2)  | -0.48 | 125   | -0.355(2)  | 2.36(3)   | -10(3)    |
|        | 11.11 | 29    | 0.4869(7) | 0.480(2)  | 0.35(1)  | -0.69 | 125   | -0.416(3)  | 2.93(6)   | -23(5)    |
|        | 9.33  | 29    | 0.4204(5) | 0.524(1)  | 0.33(1)  | -0.88 | 125   | -0.488(4)  | 3.9(1)    | -53(10)   |
|        | 7.73  | 29    | 0.3533(4) | 0.573(1)  | 0.30(1)  | -1.00 | 125   | -0.543(5)  | 4.9(2)    | -97(17)   |
|        | 6.30  | 29    | 0.2854(3) | 0.6308(8) | 0.273(8) |       |       |            |           |           |

Table 11: Fitting results of the nested-path correlation function $\mathcal{W}_k(L)$ for $SUJ_4$ and $SUJ_8$ by Eq. (7).