# Peer review of "Universality of closed nested paths in two-dimensional percolation"

_SciPost Physics_

## Round 1 · Referee Report · Anonymous (Referee 1) · 2024-5-18

Strengths

1) The present work studies a natural operator for 2D percolation, defined in terms of nested paths. The questions studied are both original and well motivated. 2) Several interesting properties are established or observed. This should stimulate further investigation (and it already has, see my comment below). 3) The numerical work is performed very carefully, providing remarkably accurate results for various questions.

Weaknesses

n/a

Report

This paper builds on earlier work [34] to study a natural operator for critical percolation in dimension two, defined in terms of nested paths. The authors both develop a better theoretical understanding, and perform thorough simulations.

In particular: 1) They check universality of the critical behavior for the nested-path operator. 2) They point out an interesting (and somewhat unexpected) behavior around the special value $k_c = -1$, which seems to display properties which are reminiscent of a BKT transition. However, there does not seem to be any theoretical reason for this at the moment, and it would certainly be interesting to examine this regime further. 3) Finally, the individual probabilities $\mathbb{P}_\ell$ that there exist $\ell$ disjoint closed nested paths, $\ell \geq 0$, are analyzed.

The paper is written very carefully, and raises intriguing questions. I strongly recommend it for publication.

Requested changes

I do not have any substantial request, since as I said, the paper is well written. I would only suggest to mention the recent preprint "Boundary touching probability and nested-path exponent for non-simple CLE" by Morris Ang, Xin Sun, Pu Yu, Zijie Zhuang (https://arxiv.org/abs/2401.15904), which was posted after the present paper was submitted (and was actually inspired in part by it, see its abstract).

l.395: For clarity, I suggest to add "exactly" $l$ closed...

I also noted a few minor typos: "mostly probably" (l. 482), "enter" (l. 507), "respective" (l. 557).

Recommendation

Publish (surpasses expectations and criteria for this Journal; among top 10%)

---

## Round 1 · Referee Report · Anonymous (Referee 2) · 2024-7-10

Strengths

1) The questions studied provide testing grounds for a number of different approaches to critical phenomena, in particular in 2D.

2) Combining theoretical reasoning with numerical work, the authors suggest some interesting relations. These in some cases can be proven rigorously while in other cases tested numerically, and in the process strengthen our grasp of universality

3) The authors take some care to be clear as to which parts of the argument are rigorously sustainable, which are conjectured based on theoretical reasoning, and which form an extrapolation of numerical work.

4) The subject has been the arena for rigorous studies from the bottom up (i.e. lattice perspective) and from up down, i.e. conformal field theory. The authors refer to a number of such studies.

Weaknesses

More could perhaps be said in terms of links of this work with the other perspectives indicated above. But it is understandable that doing so would risk clouding the author's clearly stated narrative.

Report

I definitely recommend this work for publication. It will be a stimulating contribution to an active subject.

Recommendation

Publish (easily meets expectations and criteria for this Journal; among top 50%)

---

## Editorial Decision

resubmitted